# Transcription and translation contribute to gene locus relocation to the nucleoid periphery in *E. coli*

Sora Yang[1,5], Seunghyeon Kim[2,5], Dong-Kyun Kim [3], Hyeong Jeon An[2], Jung Bae Son[1], Arvid Hedén Gynnå[4] & Nam Ki Lee [1]*

Transcription by RNA polymerase (RNAP) is coupled with translation in bacteria. Here, we observe the dynamics of transcription and subcellular localization of a specific gene locus (encoding a non-membrane protein) in living *E. coli* cells at subdiffraction-limit resolution. The movement of the gene locus to the nucleoid periphery correlates with transcription, driven by either *E. coli* RNAP or T7 RNAP, and the effect is potentiated by translation.

[1] Department of Chemistry, Seoul National University, Seoul 08826, Korea. [2] Department of Physics, Pohang University of Science and Technology, Pohang 37673, Korea. [3] School of Interdisciplinary Bioscience and Bioengineering, Pohang University of Science and Technology, Pohang 37673, Korea. [4] Department of Cell and Molecular Biology, Uppsala University, 75236 Uppsala, Sweden. [5]These authors contributed equally: Sora Yang, Seunghyeon Kim. *email: namkilee@snu.ac.kr

Transcription, the process by which mRNA is generated from DNA, is one of the most fundamental processes in cells and has been extensively studied using X-ray crystallography[1,2], bulk biochemical assays[3], and single-molecule experiments[4–10]. Much of the knowledge of transcription has been derived from in vitro studies, in which purified RNA polymerases (RNAPs) and relatively short DNAs have been used at dilute concentrations in isolation from other machinery proteins. However, the environment in a living cell is quite different from in vitro conditions. The complex structure of the nucleoid and the active reactions of other proteins, such as DNA polymerases, other RNAPs, and ribosomes, on the same DNA and mRNA molecules may affect transcription reactions in living cells[11]. One interesting example of these effects is that transcription is highly coupled with translation in that RNAP is followed immediately by ribosomes[12]. Recent studies have shown that transcription–translation coupling is more complicated than previously thought; the transcriptional rate is determined by the translational rate via direct interactions between RNAP and ribosomes[13–18]. However, fluorescence imaging studies have shown that RNAPs and ribosomes are located mainly in different subcellular regions in bacterial cells, i.e., RNAPs localize to the nucleoid area, while ribosomes localize outside nucleoids[19,20]. Thus, how transcription and translation are spatially coupled in bacterial cells remains unclear[21,22]. It is well known that *Escherichia coli* (*E. coli*) RNAPs form distinct foci located at the nucleoid periphery under fast growth conditions[20,22,23]. These foci have been suggested to actively transcribe genes, such as *rrn* operons[20,23,24]. However, it has not been directly proven whether gene loci located at the nucleoid periphery are highly transcribed by RNAPs or gene loci located inside the nucleoid move to the nucleoid periphery due to transcription[23].

Transcription in living cells has been investigated by visualizing single mRNA molecules[25,26] and detecting mRNA production in real time[27,28]. In addition, the localization[20,24,29] and tracking[23,30] of RNAPs were achieved using fluorescent protein (FP)-tagged RNAPs in live *E. coli* cells. To study in vivo transcription at the single-protein level, RNAP fused with FP has to be maintained at a low copy number[31–33]. More importantly, the *E. coli* genome contains thousands of promoters to which RNAPs can bind, which makes it nearly impossible to visualize the location of a specific gene locus occupied by RNAPs[23].

Here, we report the direct observation that the subcellular location of a non-membrane protein's gene locus under transcription moves to the nucleoid periphery (or toward the plasma membrane), an area known as the ribosome-rich region. To overcome the limitation of using FP-tagged endogenous *E. coli* RNAP, we use T7 RNAP fused with enhanced yellow fluorescent protein (eYFP) and inserted one copy of the T7 RNAP-specific promotor in the *E. coli* genome[34]. Using this system, we directly observe T7 RNAP actively transcribing a specific gene at the single-molecule level in living cells. We also use a method of marking the location of a gene using DNA-binding proteins[35,36] to observe the location of a specific gene locus during transcription by *E. coli* RNAP. We show that the subcellular relocation of gene loci is a general phenomenon that occurs during transcription by both *E. coli* RNAP and T7 RNAP in *E. coli* cells. Using T7 RNAP transcription system that is uncoupled with translation, we demonstrate that two factors are involved in gene locus movement during transcription and responsible for the relocation of gene loci during transcription to the nucleoid periphery: transcriptional activity and ribosome binding to mRNAs. When transcription by endogenous *E. coli* RNAP is coupled with translation by ribosome, however, the degree of relocation of gene locus is significantly enhanced.

## Results

**Localization of gene loci actively transcribed by T7 RNAP.** In order to visualize the location of a specific gene locus occupied by RNAPs, we replaced the endogenous promoter of the *lac* operon with a T7 RNAP-specific promoter (Fig. 1a). T7 RNAP is a single-subunit enzyme that performs all transcription reactions, like multisubunit RNAPs[1,37,38]. T7 RNAP was expressed from the plasmid pNL003 (inducer, L-rhamnose). We achieved an expression level of ~35 copies of eYFP-T7 RNAP to detect diffraction-limited spots in a cell (Supplementary Fig. 1). Two lac operators (*O1*) were placed up- and downstream of the T7 promoter[31], which tightly suppressed expression by T7 RNAP in the absence of the inducer isopropyl-β-D-1-thiogalactopyranoside (IPTG) (Fig. 1a). No localized fluorescent spots of eYFP-T7 RNAP were detected in the absence of IPTG, i.e., transcription was blocked (Fig. 1b). However, after the addition of 1 mM IPTG, which induces the dissociation of LacI from the operator sites, eYFP-T7 RNAPs were able to bind the T7 promoters and transcribe downstream of the promoters. Indeed, T7 RNAPs actively transcribing the gene were detected as diffraction-limited spots in a cell (Fig. 1c). Each cell typically contained one or two foci depending on its cell cycle. This new transcription imaging system allowed us to probe the subcellular locations of only actively transcribing RNAPs and a specific gene locus in a living *E. coli* cell.

**Direct measurement of the elongation rate of T7 RNAP in cell.** The diffraction-limited fluorescent spots indicated the location of T7 RNAPs transcribing downstream of the inserted T7 promoter (Fig. 1c). To confirm that the fluorescent foci shown in Fig. 1c indicate the locations of genes actively transcribed by T7 RNAPs, we measured the elongation kinetics of T7 RNAP using fluorescent imaging. A simple kinetic model was used to describe the transcriptional kinetics (Fig. 1d); $k_{on}$ denotes the transcriptional on-rate, which includes promoter searching by RNAPs, RNAP-promoter complex formation, and transcription initiation, and $k_{off}$ is the transcriptional off-rate, which is determined by the elongation duration time, i.e., the time an RNAP spends as it moves from the promoter site to the terminator site. These transcriptional kinetic parameters can be derived from the appearance of fluorescent foci and an increase in their intensity, which represents the number of T7 RNAPs transcribing genes per cell as a function of time after IPTG addition (Fig. 1e and Supplementary Note 1). Gene length was controlled by inserting tandem T7-specific termination sites downstream of *lacY* (Fig. 1a), which resulted in a gene length of ~4.5 kilo-base pairs (kbps) (strain T7p_4.5kb). We fit the data of T7p_4.5kb with a 4.5-kbp gene length using the kinetic model (Fig. 1e, blue) and obtained $k_{on}$ ($0.46 \pm 0.03$ min$^{-1}$, mean ± standard deviation) and $k_{off}$ ($1.12 \pm 0.11$ min$^{-1}$). Considering that the elongation duration time, which is longer than tenths of a second, is much longer than the transcription termination time[39], we obtained the elongation rate ($4.5$ kbp $\times k_{off}$), which was $84 \pm 8$ bp s$^{-1}$. $1/k_{on}$, which was $131 \pm 10$ s, denotes the average time for T7 RNAPs to get into an elongation state after their repeated attempts to find the promoter and initiate transcription.

Then, as a control, we shortened the gene length to 3.3 kbp by placing termination sites at the end of the *lacZ* gene (T7p_3.3kb). We expected that the elongation rate and transcription initiation rate would be invariant, while the elongation duration time would be reduced by the shortened gene length. Indeed, the elongation time was reduced to $36 \pm 3$ s (T7p_3.3kb) from $54 \pm 6$ s (T7p_4.5kb), but the elongation rate and $1/k_{on}$, which were $92 \pm 7$ bp s$^{-1}$ and $149 \pm 14$ s, respectively (Fig. 1e, red), were similar to those of T7p_4.5kb. These results demonstrate the

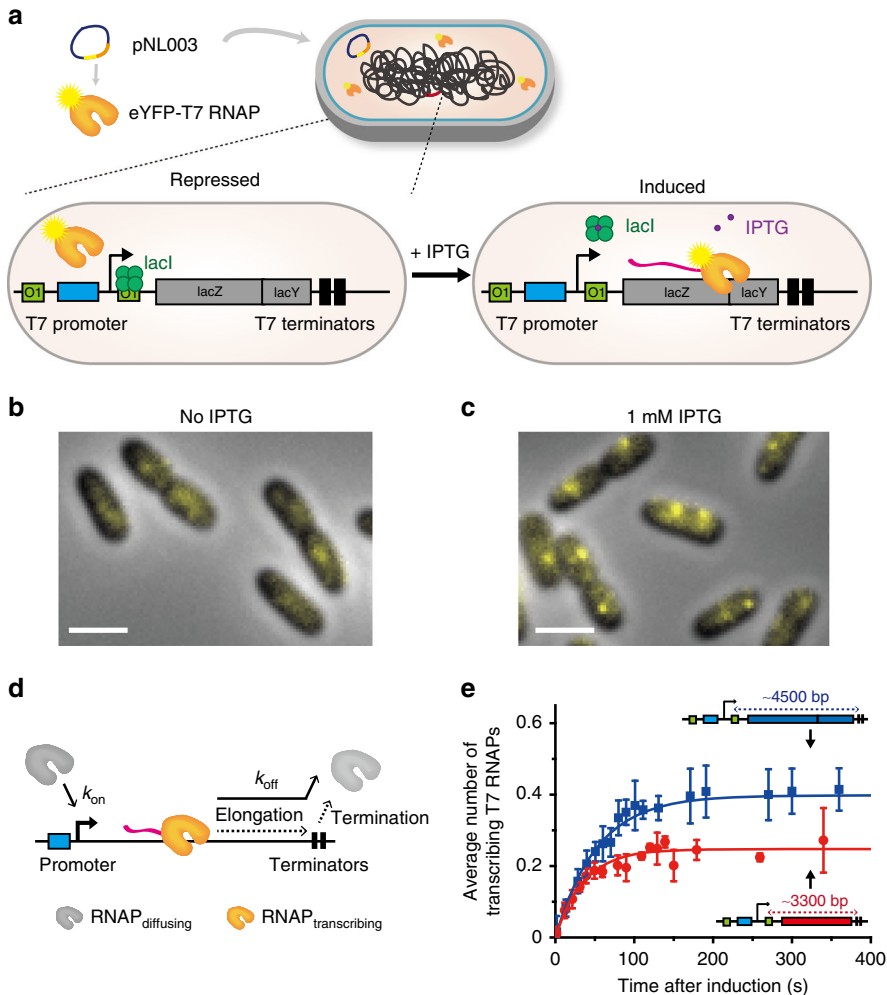

**Fig. 1** Transcription by T7 RNAP in living *E. coli* cells was observed by single-protein detection. **a** A schematic illustration of the gene system used to visualize single eYFP-T7 RNAPs during transcription. eYFP-T7 RNAP is expressed from an L-rhamnose inducible plasmid (pNL003). The left panel represents the repressed condition, in which the binding of LacI proteins on two Lac operators (O1) blocks transcription by RNAP. The right panel represents the induced condition, in which LacI proteins dissociate from their operators with the addition of 1 mM IPTG and RNAP thus binds to the T7 promoter and generates mRNAs. **b**, **c** Representative images of eYFP-T7 RNAPs in *E. coli* cells. Scale bar, 2 μm. **b** No IPTG. Blurred fluorescent signals from the rapidly diffusing eYFP-T7 RNAP molecules were detected. **c** With 1 mM IPTG. The image was acquired at 303 s after adding IPTG. Diffraction-limited fluorescent foci of eYFP-T7 RNAP were clearly observed. **d** A kinetic model of in vivo transcription. $k_{on}$ denotes the transcriptional on-rate, and $k_{off}$ denotes the transcriptional off-rate (Supplementary Note 1). **e** Average number of transcribing eYFP-T7 RNAPs during transcription per cell after IPTG induction, including cells that exhibit no bright spots. The blue filled squares and the red filled circles denote data from the induction of the T7p_4.5 kb and T7p_3.3 kb strains, respectively. The data were fitted to a single exponential function, $a(1-\exp(-bt))$, where $a = k'_{on} \times [\text{RNAP}_{total}]/k_{off}$ and $b = k_{off}$. The elongation rate of RNAP equals $L_{gene\ length} \times k_{off}$. The average total number of T7 RNAPs, $[\text{RNAP}_{total}]$, in a cell was 35. The data represent mean ± s.d. (standard deviation) obtained from three independent experiments

robustness of our measurement of the transcription on-rate ($k_{on}$) and off-rate ($k_{off}$), i.e., the elongation rate. To reconfirm the elongation rates measured by image analysis shown in Fig. 1d, e, we determined the elongation rate based on real-time reverse-transcription polymerase chain reaction (RT-PCR). To probe the elongation rates in both the T7p_4.5kb and T7p_3.3kb strains, we designed two TaqMan probes to detect the fluorescent signals from probes complementary to the segments of the *lacZ* transcript (Supplementary Fig. 2). The elongation rates measured by RT-PCR were 101 ± 7 and 91 ± 12 bp s⁻¹ for the T7p_4.5kb and T7p_3.3kb strains, respectively, which are consistent with the results of image analysis shown in Fig. 1. It is possible that the average number of transcribing T7 RNAPs in Fig. 1e was underestimated due to the photobleaching effect.

These results confirm that the fluorescent foci in Fig. 1c indicate T7 RNAPs actively transcribing genes. The time-

dependent analysis of T7 RNAP fluorescent foci provides an excellent method to probe transcriptional kinetics in living cells. We determined that the elongation rate of T7 RNAP in live *E. coli* cells in a M9 minimal medium at room temperature is ~90 bp s⁻¹.

**Transcription by T7 RNAP induces the relocation of gene loci.** Next, we analyzed the localization of transcribing genes using fluorescent foci of T7 RNAP (Fig. 2). A bacterial chromosome is highly condensed and forms a unique structure, the nucleoid, in the center of the cell[40], as the genome is compacted to fit into a small occupied region in an *E. coli* cell[36,41]. Thus, the subcellular positions of eYFP-T7 RNAP foci were expected to be mainly localized at the central region of the cell. Interestingly, we often observed localization of the fluorescent foci close to the plasma

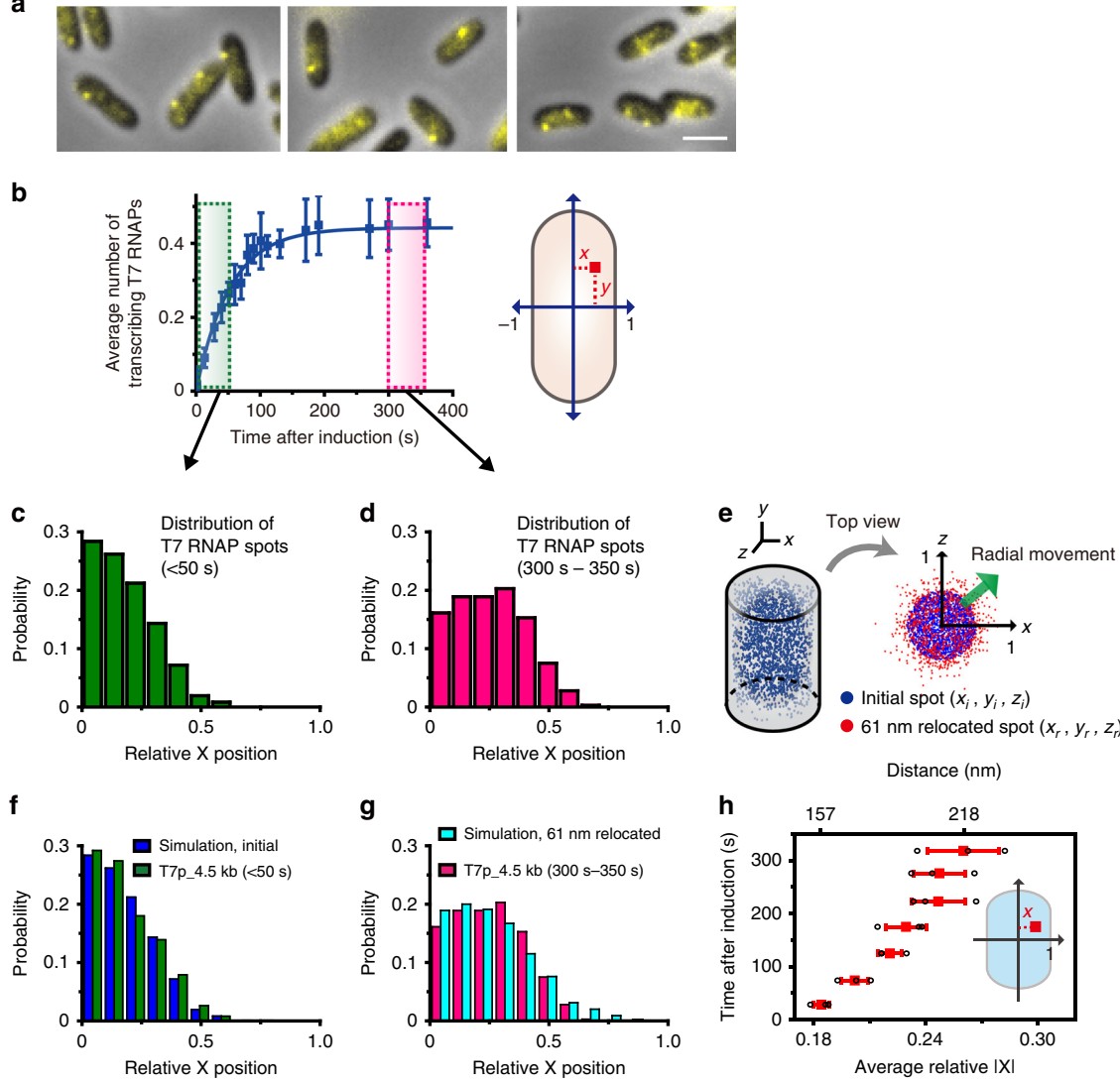

**Fig. 2** Gene locus moves to the nucleoid periphery by T7 RNAP-driven transcription. **a** Representative images of T7p_4.5kb cells acquired late (>270 s) after adding IPTG. Fluorescent foci denote the locations of gene loci actively transcribed by T7 RNAP. Scale bar, 2 μm. **b** Analysis of the subcellular localization of gene locus under transcription. Left panel of **b** is a reproduction of Fig. 1e, blue line, for convenience. The green box indicates the time window for the cells at the initial transcription stage, <50 s. The purple box indicates the time window for the cells between 300 and 350 s after IPTG induction. The right panel in **b** shows normalization of the cell size. The location of each fluorescent spot was determined as relative coordinates (x, y) to take into account differences in cell size. The x-axis corresponds to the short axis of the cell. **c, d** Distributions of the subcellular localization of transcribing eYFP-T7 RNAP foci along the short axis. **c** The distribution of the locations of transcribing RNAP spots within 50 s after induction (total of 363 fluorescent foci). **d** The distribution of the locations of transcribing RNAP spots between 300 and 350 s after induction (total of 360 fluorescent foci). **e** Simulation of gene loci movements. A total of 100,000 random spots with a 70-nm localization error were generated in the cylindrical coordinates (blue spots). Red spots indicate the final locations of each spot randomly moved in the radial direction by 61 nm on average. **f, g** Comparison of the simulated and experimental results. **f** The distribution obtained by the simulation (blue bar) was similar to the distribution of the initial transcription stage (green bar, same as Fig. 2c). **g** The spots obtained by the simulation in Fig. 2e were randomly moved in the radial direction by 61 nm on average (cyan bar). The distribution following these movements is comparable with the distribution in Fig. 2d (purple bar). **h** The average relative position of the transcribing gene loci after induction. The distance in 3D geometry is presented. Data represent mean ± s.d. obtained from three independent experiments (red). Each point represents independent measurement (black)

membrane (Fig. 2a and Supplementary Fig. 3). Thus, we analyzed the locations of the fluorescent foci in each cell over time after inducing transcription with IPTG (Fig. 2b–d). Typical two-dimensional (2D) Gaussian fitting was used to determine the locations of fluorescent foci, which resulted in an ~70-nm localization resolution in our measurements (Supplementary Fig. 4a). Then, the normalized positions (x, y) of the foci in the 2D projection relative to the short and long axes of each cell, respectively, were determined to account for size differences between

cells (Fig. 2b). Although the localization error of each dot was 70 nm, the error of the averaged subcellular localization was lower than 30 nm, as the number of measured foci increased to 350 at the 5% significance level (Supplementary Fig. 4b–d).

To determine whether the spatial localization of actively transcribing genes varies, we obtained the distributions of the relative positions of foci in the x-axis within different time windows after transcription induction (Fig. 2b–d). Figure 2c shows the distribution of transcription foci appearing within 50 s

after induction. This distribution is similar to the distribution of FP-labeled LacI (Supplementary Fig. 5), which denotes the locations of the lac operon without transcription (repressed condition)[31]. Interestingly, transcription foci that appeared between 300 and 350 s after induction moved to the nucleoid periphery (radial movement), and their distribution was significantly different compared with their initial distribution (Fig. 2d). The difference in the average position on the relative x-axis between the initial (<50 s) and final conditions (300–350 s) was 0.07 (normalized value), which corresponds to 61 nm in 3D geometry (Supplementary Note 2).

To test whether a movement of 61 nm to the nucleoid periphery induced the differences in transcription distribution that we observed, we performed a simulation (Fig. 2e–g); we randomly generated 100,000 spots with boundary conditions of $-l < x_i, y_i < l$ and a radial distance ($r = \sqrt{x_i^2 + y_i^2}) < l$ to mimic nucleoid condensation and gave a 70-nm localization error for each spot (Fig. 2e). When $l$ equaled 0.35, the simulated distribution in the x-axis was very similar to that in Fig. 2c, which shows the initial transcription distribution (Fig. 2f). Then, we randomly moved the spots in the radial direction by an average of 61 nm (Supplementary Note 2). The relocated distribution determined by the simulation matched well with the distribution shown in Fig. 2d (Fig. 2g). As shown in Fig. 2h, the average relative position of transcribing genes along the x-axis increased steadily over time, which indicates that the gene locus moved to the nucleoid periphery due to active transcription by T7 RNAP. Our results imply that transcription is initiated within the nucleoid and then the gene locus being transcribed by T7 RNAP moves to the nucleoid periphery.

Because we placed tandem transcription terminator sites in the 4.5-kbp region, the T7p_4.5kb strain expresses both LacZ and LacY. Thus, it is possible that the transertion of LacY, i.e., the cotranscriptional translation and simultaneous insertion of membrane proteins in the plasma membrane[42], causes repositioning of the chromosomal locus. To remove the effect of transertion on gene movement, we used the T7p_3.3kb strain, which contains only lacZ. Again, we observed the repositioning of the transcribed gene locus in the T7p_3.3kb strain (Supplementary Fig. 6), which was nearly identical to the results obtained in the T7p_4.5kb strain (Supplementary Fig. 7). This result indicates that the gene movement was not mainly caused by the expression of the membrane protein LacY in our observation time window.

**Movement of a non-membrane protein gene loci by E. coli RNAP**. Our observation that transcription by T7 RNAP causes the movement of gene loci immediately raises the question of whether this movement also occurs with transcription by endogenous E. coli RNAP. Endogenous E. coli RNAP transcribes ~4000 genes. Thus, it is impossible to identify the location of a specific gene by using fluorescence-labeled E. coli RNAP[23]. We used an alternative method of marking the location of a gene using DNA-binding proteins[35,36]. We incorporated six tandem TetO sites (6xTetO) downstream of the lacZ gene in an E. coli chromosome (strain lacZ-6xTetO) without the membrane protein lacY gene. The TetO-binding protein TetR fused with eYFP (TetR-eYFP) was expressed from an inducible plasmid (Fig. 3a). We detected diffraction-limited TetR-eYFP spots in cells with a localization error of ~30 nm, allowing localization of the lacZ genes (Fig. 3a). The localized TetR-eYFP foci without the 6xTetO site in the genome were not observed (Supplementary Fig. 8). Then, the relative positions of the TetR-eYFP foci were determined along the short axis (x-axis) of E. coli cells, as performed in Fig. 2. The time-dependent averaged locations of the fluorescent foci were plotted after transcription was induced with 1 mM

IPTG (Fig. 3b, c). The average locations of the foci moved to the nucleoid periphery (Fig. 3b, blue, Supplementary Movies 1 and 2). Then, we replaced lacZ with the mCherry gene to check that the movement was not lacZ gene-specific (Fig. 3c). The mCherry gene (which encodes a non-membrane protein) also moved to the nucleoid periphery by transcription of E. coli RNAP (Fig. 3c). Thus, movement of gene loci to the nucleoid periphery generally occurs by both T7 RNAP- and E. coli RNAP-driven transcription.

**Effect of translation on gene locus movement**. What factor is responsible for the observed movement of gene loci during transcription? The transertion of a membrane protein was not the major factor underlying this movement (Supplementary Fig. 7). Because ribosomes are typically excluded from the nucleoid[43], the formation of a large DNA-RNAP-mRNA-ribosome complex may induce the movement of gene loci during transcription. To test this possibility, we blocked ribosome binding to mRNA and the coupling between transcription and translation by deleting the ribosomal binding site (RBS) from the lacZ gene (lacZ-6xTetO_ΔRBS) (Fig. 3b, d). We used E. coli RNAP-driven transcription in this study because transcription by E. coli RNAP is coupled with translation while T7 RNAP transcription is decoupled with translation. RBS-deleted strains expressed several hundred times lower levels of lacZ proteins than a control strain (Miller assays, Supplementary Fig. 9). Interestingly, relocation of the gene locus during transcription was dramatically reduced in the RBS-deleted strain (Fig. 3b, gray); the movement of the lacZ gene locus with the RBS was ~32 ± 9 nm, while nearly no movement was observed for the lacZ gene locus without the RBS (2 ± 4 nm) (Fig. 3d). When we inhibited translation initiation using kasugamycin (Ksg) treatment[44], no movement of the gene locus after transcription induction was observed (Supplementary Fig. 10). This result is consistent with the results in the RBS-deletion strain (lacZ-6xTetO_ΔRBS) (Fig. 3d). Then, we enhanced the gene expression level by increasing the temperature to 37°C (Fig. 3e). As the expression level increased, the distance moved by the gene locus with the RBS (lacZ-6xTetO) increased to 48 ± 7 nm (Fig. 3e, blue bars). Again, when the RBS was deleted (lacZ-6xTetO_ΔRBS), the distance moved by the gene locus was dramatically reduced to 18 ± 11 nm (Fig. 3e, gray bar). Then, to completely abolish translation, we removed the start codon from the lacZ mRNA (Fig. 4a) and found that the strain showed stronger translation inhibition than that in the RBS-deleted strain (Fig. 4b). As expected from the decreased translation activity, we did not observe gene movement in the strain, in which the start codon from the LacZ mRNA had been deleted, at 37°C (Fig. 4c). The degree of gene movement was strongly correlated with translation activity (Fig. 4d). These results show that translation contributes to the subcellular relocation of gene loci under transcription in E. coli.

**Gene loci movement depends on the transcriptional activity**. Distinct foci, formed by E. coli RNAPs and located at the nucleoid periphery under rapid growth conditions, have been suggested to be highly transcribed genes[20,22,23]. It is still unclear whether the gene loci located at the nucleoid periphery are highly expressed by RNAP or whether the gene loci located inside the nucleoid move to the nucleoid periphery during transcription. In addition, it is unknown whether the distance of the movement of a specific gene locus to the nucleoid periphery depends on transcriptional activity[23]. Our temperature-dependent measurements demonstrated that movement of the gene locus to the nucleoid periphery increases as transcriptional activity increases (Fig. 3d, e). To further support this observation, we constructed a strain that expressed LacZ more actively (Fig. 5a). The T7 RNAP

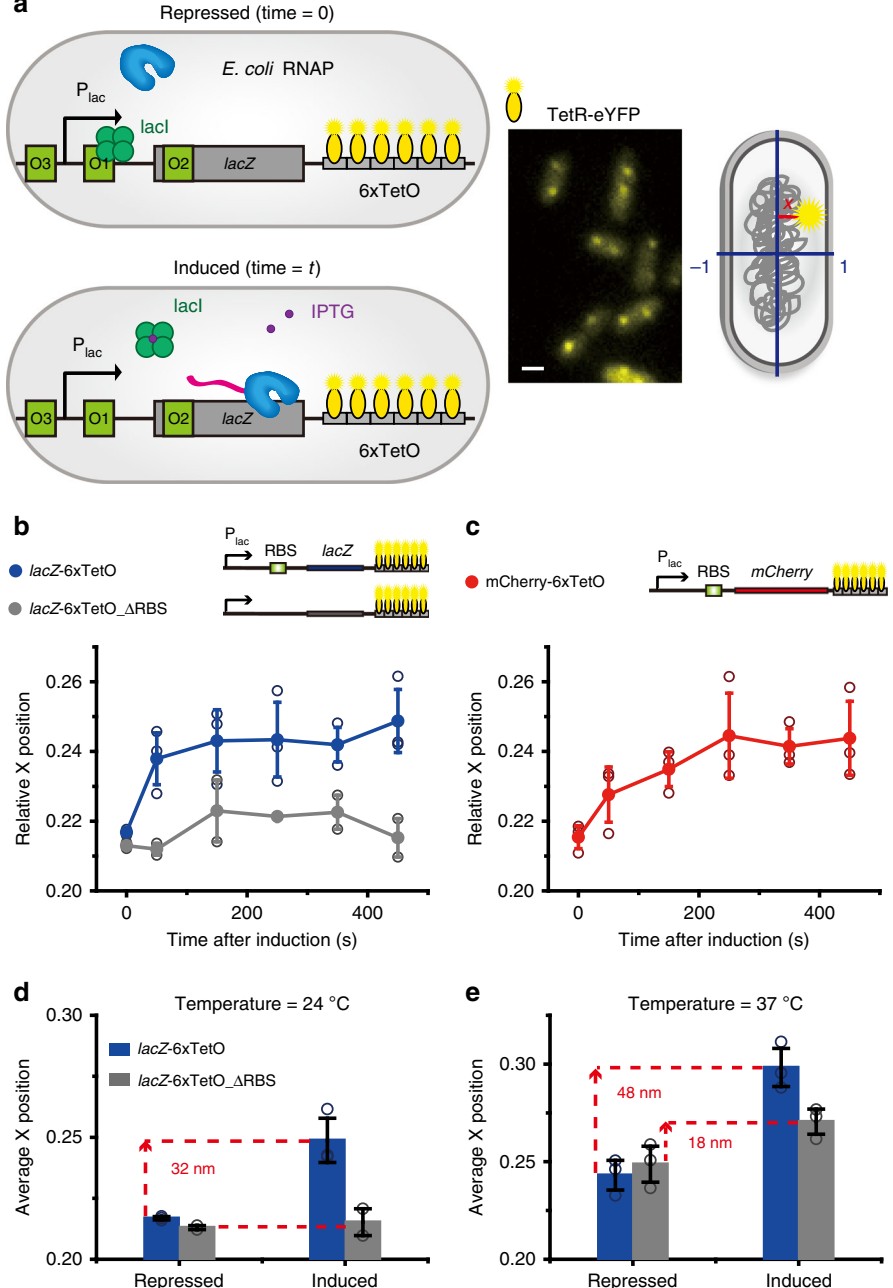

**Fig. 3** Direct observation of the movement of a non-membrane protein gene locus by transcription of *E. coli* RNAP. **a** A schematic of the gene system used to detect location of the *lacZ* gene locus transcribed by *E. coli* RNAP (left panel). Six repeats of TetO (6xTetO) were inserted downstream of the *lacZ* gene or *mCherry* gene transcribed by *E. coli* RNAP. TetR-eYFPs bound to the TetO array were detected as a fluorescent spot (middle panel). The localization error was 30 nm. Scale bar, 1 μm. **b** Quantitative analysis of the movement of the *lacZ* gene locus. The *lacZ* gene locus moved to the nucleoid periphery after IPTG induction (*lacZ*-6xTetO strain, blue squares) (>1500 spots). Movement of the *lacZ* gene locus following the RBS deletion (*lacZ*-6xTetO_ΔRBS strain, gray circles) was not observed (>800 spots). Data represent mean ± s.d. obtained from three independent experiments. Each point represents independent measurement. **c** Quantitative analysis of *mCherry* gene locus movement. Movement of the *mCherry* gene to the nucleoid periphery was observed after IPTG induction (*mCherry*-6xTetO, red squares) (>940 spots). Data represent mean ± s.d. obtained from three independent experiments. Each point represents independent measurement. **d**, **e** Comparison of gene locus movement with and without the RBS at 24 °C (**d**) and 37 °C (**e**). Average relative x-positions of gene loci were obtained without IPTG (repressed) and 5 min after adding IPTG (induced). Blue bar, *lacZ*-6xTetO, and gray bar, *lacZ*-6xTetO_ΔRBS. Bar and error bars represent mean ± s.d. from three independent experiments. Each point represents independent measurement

transcription system allows us to control the transcriptional activity through T7 RNAP concentration variation. Thus, we used T7 RNAP transcription system in this study. High transcription levels were achieved by using high concentrations of T7 RNAP, and the gene location was detected by 12xTetO (T7p_*lacZ*-12xTetO). Indeed, the distance moved by the *lacZ* gene locus was

significantly increased to 81 ± 14 nm at 24°C (Fig. 5b–d) relative to the 32 ± 9 nm distance driven by *E. coli* RNAP, as shown in Fig. 3d. In addition, the outward movement of the *lacZ* gene locus in the axial direction (78 ± 32 nm) was also observed in the T7p_*lacZ*-12xTetO strain (Supplementary Fig. 11), which was not observed in other experimental conditions that expressed less

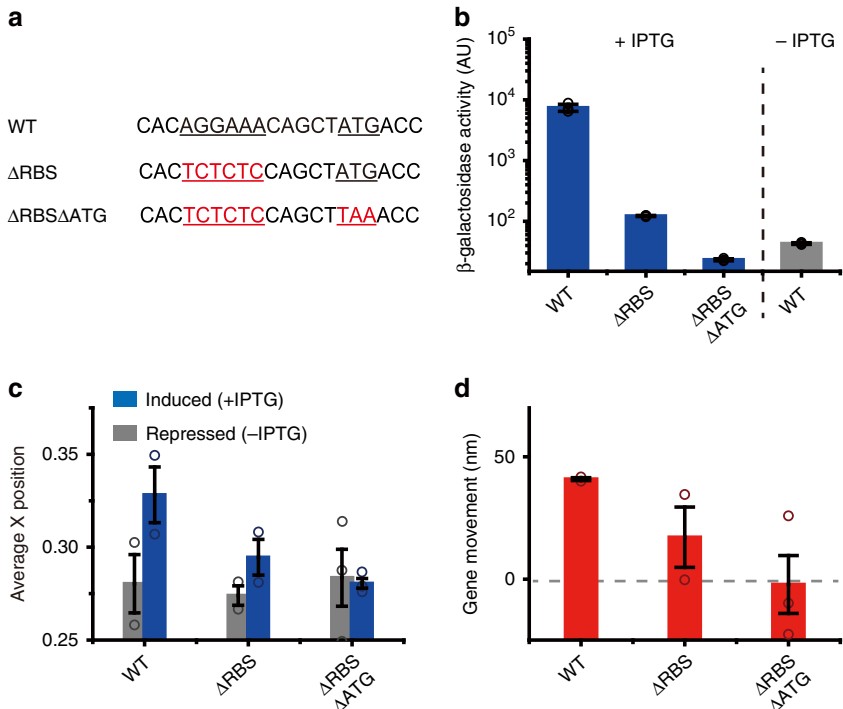

**Fig. 4** Translation-dependent gene loci movement toward the nucleoid periphery in *E. coli* RNAP transcription. **a** 5′ UTR sequence of the mutated strains. The underlined AGGAAA in the WT strain (*lacZ*-12xTetO) was mutated to TCTCTC (red letters) (ΔRBS, *lacZ*-12xTetO_ΔRBS) for RBS substitution. The start codon of the *lacZ* gene (underlined) was mutated to TAA (ΔRBSΔATG, *lacZ*-12xTetO_ΔRBS_ΔATG). **b** β-galactosidase assay showing that LacZ expression in the strain in which the start codon in the ΔRBS strain was replaced by a stop codon (ΔRBSΔATG) decreased to ~20% of that in the ΔRBS strain (blue bars, 1 mM IPTG induction of each strain). The LacZ expression level in ΔRBSΔATG cells was lower than that in WT cells under repressed conditions (no IPTG, gray bar). **c** Average x-positions of the *lacZ* gene locus were determined in the absence (repressed, gray bars) and presence of IPTG (induced, blue bars). **d** The degree of gene movement is the difference in the average position before (repressed) and after (induced) induction in **c**. Bar and error bars represent mean ± s.d. from three independent experiments. Each point represents independent measurement

amounts of mRNAs compared with T7p_*lacZ*-12xTetO strain. This result clearly shows that the transcription reaction induces the relocation of gene loci and that the degree of this movement depends on transcriptional activity.

Then, we measured the extent of gene locus movement relative to the size of the nucleoid. We imaged the nucleoid using SYTOX-Green dye[45]. The nucleoid width and length were defined as the full-width at half-maximum of the intensity profiles of SYTOX fluorescent images, as described by Bakshi et al.[45]. (Supplementary Fig. 12). The size of the nucleoids was maintained for the various measurements using both T7 RNAP and *E. coli* RNAP (short axis = 0.60 ± 0.01 μm). Thus, the width of the nucleoid from the center of the cell was 300 nm. As a result, gene movements during *E. coli* RNAP transcription (30–50 nm) and T7 RNAP transcription (60–80 nm) correspond to 10–16% and 20–27% of the nucleoid size (width), respectively.

**Analysis of the factors contributing to gene loci movement**. We showed that translation enhances gene locus movement in *E. coli* RNAP-driven transcription (Figs. 3d, e and 4). We then examined how translation influences gene locus movement. We first tested whether ribosome binding to mRNA, which results in the formation of a large DNA-RNAP-mRNA-ribosome complex, induces the relocation of gene loci. To remove the effects of direct interactions between RNAPs and ribosomes[15–17] and quantify only the effect of ribosome binding to mRNAs to the degree of gene loci movement, we used T7 RNAP-driven transcription for this analysis (T7p_*lacZ*-12xTetO). T7 RNAP has no known interaction with ribosomes and moves approximately five times faster than *E. coli* RNAP. We blocked ribosome binding to the

DNA–T7RNAP–mRNA complex by adding Ksg, which inhibits translation initiation[44]. The blocking of translation initiation resulted in a reduced gene locus movement of 59 ± 2 nm (Fig. 5e and Supplementary Fig. 13), which is ~30% less than the movement observed without Ksg (Fig. 5d). Since it has been known that Ksg treatment leads to the change in the nucleoid morphology[44], we assessed whether the nucleoid morphology before and after Ksg treatment differ and observed no change in the nucleoid morphology in our experimental conditions (Supplementary Fig. 14). To confirm this result without using antibiotics, we constructed RBS-deleted strains for T7 RNAP-driven transcription (Fig. 5f). The deletion of the RBS from the *lacZ* gene (T7p_4.5kbΔRBS and T7p_3.3kbΔRBS) also resulted in an ~35% shorter gene locus movement in both strains (Fig. 5f). These results demonstrate that ribosomal translation of mRNA without transcription–translation coupling contributes to the subcellular relocation of genes undergoing transcription. As for T7 RNAP-mediated transcription, transcriptional activity is the major factor driving gene locus movement. Considering that the spots we observed contain mostly single T7 RNAP (Fig. 1e), one DNA–T7RNAP–mRNA–ribosome complex seems to be required to drive movement of the transcribing gene locus toward the nucleoid periphery.

Although ribosome binding to mRNA increases movement by 30–35% in the T7 RNAP-driven system, this finding does not explain the effect of translation on the *E. coli* RNAP system in which movement increased significantly, as shown in Fig. 4. It has been reported that transcription activity of *E. coli* RNAP increases in the presence of the ribosomal protein S1 in vitro[46]. Thus, we suspected that transcription–translation coupling changes the

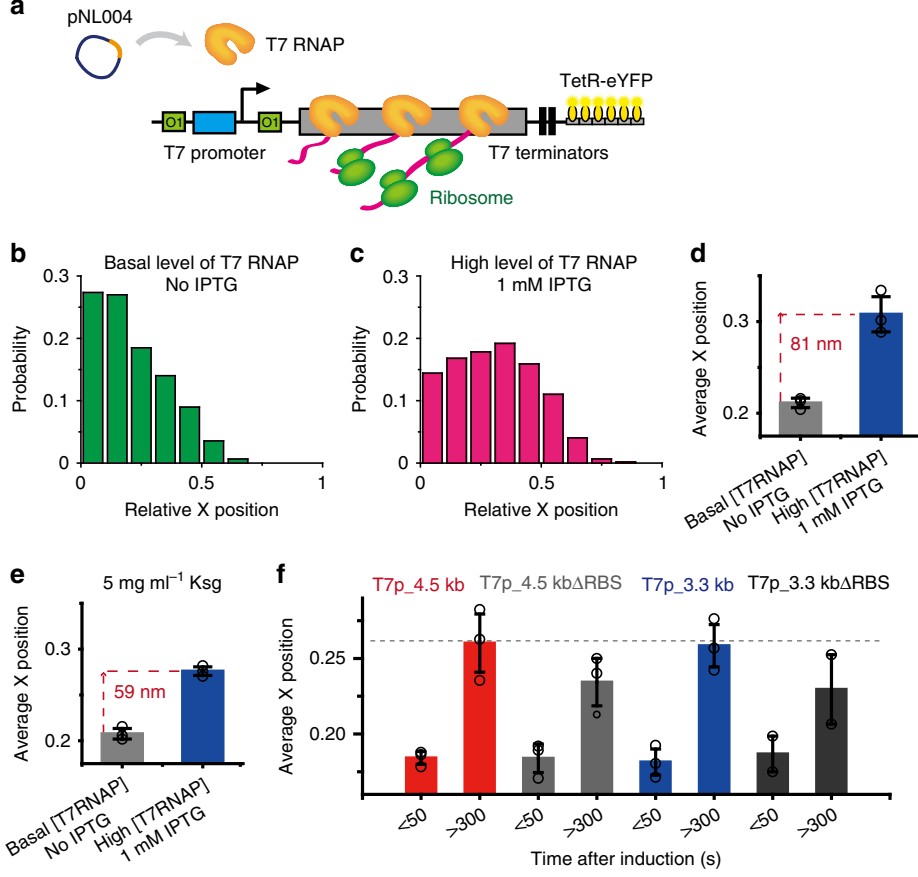

**Fig. 5** Effects of transcriptional activity and ribosome binding to mRNA on gene locus movement. **a** Schematic of the gene system used to detect the location of the *lacZ* gene transcribed by wild-type T7 RNAP. Twelve repeats of TetO were inserted downstream of tandem T7 terminators. TetR-eYFP bound to the TetO array was detected as a fluorescent spot. T7 RNAP was expressed from an L-rhamnose inducible plasmid (pNL004). **b**, **c** Distributions of the subcellular localization of a gene locus transcribed by T7 RNAP. **b** The distribution of the gene locus with a basal level of T7 RNAP (no rhamnose) and no IPTG. **c** The distribution of the gene locus when T7 RNAP was expressed by the addition of 0.2% L-rhamnose and 1 mM IPTG. **d** Comparison of gene locus movement. Average relative x-positions of the gene loci from Fig. 5b (gray bar, basal level of T7 RNAP and no IPTG) and Fig. 5c (blue bar, high expression of T7 RNAP and 1 mM IPTG). **e** Comparison of gene locus movement under translation initiation inhibition conditions. The same experiment was performed as shown in Fig. 5d after a 15-min incubation with 5 mg mL−1 kasugamycin (Ksg), which inhibits translation initiation. **f** Comparison of gene loci movement with and without the RBS. Red and blue bars denote the average relative x-positions of *lacZ* gene loci of T7p_4.5 kb and T7p_3.3 kb, respectively, within each time window. When the RBS was deleted, gene locus movement was reduced by ~30% (gray bars: T7p_4.5kbΔRBS and dark gray bars: T7p_3.3kbΔRBS). Bar and error bars represent mean ± s.d. from three independent experiments. Each point represents independent measurement

gene expression level in *E. coli* RNAP transcription. We measured the mRNA expression level of *E. coli* RNAP-driven strains using RT-PCR (Supplementary Fig. 15). Indeed, LacZ mRNA with the RBS was expressed at a 36-fold higher level relative to LacZ mRNA expression in the RBS-deleted strain (Supplementary Fig. 15a). Typically, the RBS deletion increases the mRNA degradation rate, which results in less mRNA at the steady state[47]. To take into account the rapid degradation of RBS-deleted mRNA, we measured the degradation rates of LacZ mRNA with and without the RBS. The half-life of LacZ mRNA was 5.5 ± 0.8 min, which was reduced to 2.4 ± 0.3 min for the RBS-deleted strain (Supplementary Fig. 15b). Even after considering the short half-life of mRNA, the expression level of mRNA with the RBS was 16-fold higher than that without the RBS. To confirm the RT-PCR results, we used fluorescence in situ hybridization (FISH) to measure mRNA levels (Supplementary Fig. 15c). Again, 24-fold more mRNA was expressed with the RBS. Thus, transcription–translation coupling increased transcriptional activity relative to the decoupled case (without the RBS). The enhanced transcriptional activity with transcription–translation coupling may have increased gene locus movement.

## Discussion

In this work, we have demonstrated that a gene locus moves to the nucleoid periphery by transcription. Gene locus movement depends on transcriptional activity, and translation coupled with transcription contributes significantly to gene locus movement in *E. coli* RNAP-driven transcription. Gene locus movement was observed in both T7 RNAP- and *E. coli* RNAP-dependent transcription, i.e., a common phenomenon of transcription in *E. coli* (see Supplementary Fig. 16 for a summary of the strains used in this work).

Transcription initiation via the binding of RNAP to a promoter occurs in the nucleoid region. A recent single-particle tracking experiment with ribosomes demonstrated that free ribosomal subunits are not excluded from the *E. coli* nucleoid[48]. This observation suggests that translation of nascent mRNA could start at any location in the nucleoid. Based on our results, we suggest that relocation of the gene locus by transcription occurs as follows (Fig. 6). The RNAP–promoter complex forms inside the nucleoid. Transcription induces relocation of the gene locus depending on transcriptional activity. As the ribosome binding sites of mRNAs are generated, free ribosomal subunits

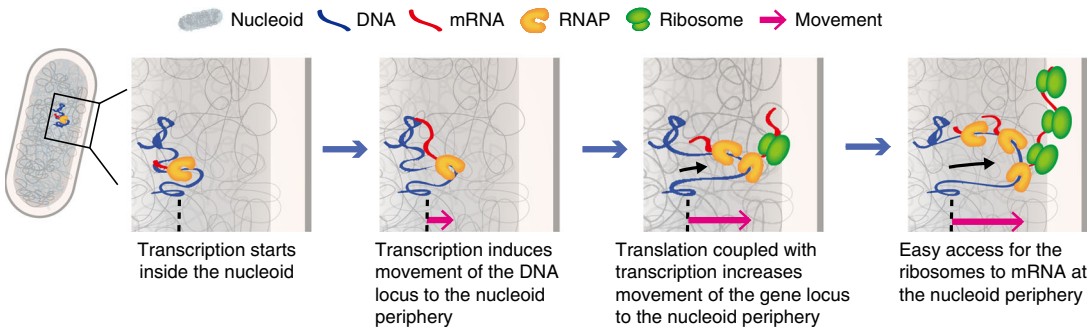

**Fig. 6** Model of gene locus movement by transcription in *E. coli* cells. Transcription starts inside the nucleoid with formation of the RNAP–promoter complex. As soon as the ribosome binding site of mRNA is generated, free ribosomal subunits in the nucleoid region bind the mRNA. The DNA–RNAP–mRNA–ribosome complex moves further outside the nucleoid for easy ribosome access

present in the nucleoid region bind the mRNAs. The DNA–RNAP–mRNA–ribosome complex moves further outside the nucleoid due to the increased transcriptional activity and ribosome binding.

Our work shows that the major factor that causes the gene loci movement during transcription is different for T7 RNAP- and *E. coli* RNAP-driven transcription. As for *E. coli* RNAP-driven transcription, translation coupled with transcription causes most gene loci movement that the deletion of both RBS site and a start codon (decoupling between transcription and translation) nearly abolishes the gene loci movement (Figs. 3 and 4). However, the transcription by T7 RNAP is not coupled with translation. The deletion of RBS site causes only 30% less movement of gene loci (Fig. 5f), which means that ribosome binding to mRNA increases the gene loci movement only 30% more. Thus, as for T7 RNAP-driven transcription, the transcriptional activity causes most gene loci movement.

Jin et al. observed foci of transcribing RNAP located at the surface of the nucleoid under fast growth conditions[20,29]. These foci were suggested to be the locations of actively transcribing genes, such as *rrn* operons. Since ribosomal RNAs of *rrn* operons are not translated by ribosomes, RNAP foci under fast growth conditions may occur due to high transcriptional activity without transcription–translation coupling. A recent live-cell super-resolution imaging study showed that highly transcribing RNAPs tend to cluster at the edge of the nucleoid in minimal media growth conditions[23]. In this case, the gene locus may have moved to the nucleoid periphery mainly by transcription-translation coupling with forming a large DNA–RNAP–mRNA–ribosome complex.

How, then, is transcriptional activity correlated with the distance of gene locus movement? The entropic force and excluded volume effect are possible sources for the movement, which have been suggested to exclude ribosomes from the nucleoid[43]. In this model, DNA polymers are confined in the center of the cell by conformational entropy, while multiple 70S-polysomes formed on nascent mRNA inside the nucleoid diffuse outside the nucleoid due to the excluded volume effect. The large DNA–RNAP–mRNA–ribosome complex may feel a strong entropic force that would drive its movement outside the nucleoid. The other possible source is that high transcriptional activity may change the local environment of the gene locus, like DNA supercoiling[49,50] and the chromosomal loop structure formed by nucleoid-associated proteins (NAPs)[51,52]. It is well known that transcription accumulates positive supercoiling ahead of RNAP and negative supercoiling behind RNAP[53]. Supercoiling due to high RNAP activity would accumulate very rapidly before timely correction by topoisomerases and gyrase, which may change the local DNA structure size range from tens

of nanometers to hundreds of nanometers[54,55]. In addition, DNA supercoiling is highly dynamic that can move by diffusion and hopping process[56], affecting several kilobases away from the origin of the supercoiling[57]. Alternatively, high RNAP activity would dissociate NAPs from DNA, maintaining the bacterial chromosomal structure, which may also change the looping and local DNA structure.

## Methods

**Strains and plasmids.** For the experiments with eYFP-T7 RNAP, *Cat* (chloramphenicol resistance gene)-O1-T7 promoter-O1 with or without *lacZ* RBS fragment was inserted into the lac promoter region of BW25993 using lambda red recombination to replace the lac promoter with the T7 promoter. T7p_4.5kb and T7p_4.5kbΔRBS strains were constructed by inserting tandem T7 terminators (Tpi)-kanamycin resistance gene (KanR) fragment into the downstream of *lacY* gene, resulting in 4.5 kb distance between the T7 promoter and T7 termination site. For the construction of T7p_3.3kb and T7p_3.3kbΔRBS strains, we put a Tpi-KanR fragment downstream of the *lacZ* gene. For the construction of the T7p_*lacZ*-12xTetO strain, a KanR-12xTetO fragment was inserted into the downstream of T7 terminators in the strain T7p_3.3kb. The *lacZ*-6xTetO strain was constructed by inserting a KanR-6xTetO fragment downstream of the *lacZ* gene, while *lacY* and *lacA* genes were deleted. The *mCherry*-6xTetO strain was constructed by replacing the *lacZ* gene with the *mCherry* gene using lambda red recombination and then a KanR-6xTetO fragment was inserted into the downstream of the *mCherry* gene. The antibiotic genes were removed using a FRT cassette after successful cloning. NL003 strain was constructed by inserting a Tsr-eYFP gene with *Cat* into the lac operon for replacing the native *lacZYA* genes. All strains are described in Supplementary Fig. 16.

T7 RNAP fused with eYFP was expressed from a low-copy-number plasmid backbone pVS133 (pNL003). L-rhamnose was used to induce eYFP-T7 RNAP expression from the plasmid, which carries the ampicillin resistant gene (ampR) for antibiotic selection.

T7 RNAP expressed from a low-copy-number plasmid backbone pVS133 (pNL004). L-rhamnose was used to induce T7 RNAP expression from the plasmid, which carries chloramphenicol resistance gene for antibiotic selection.

**Cell growth conditions.** All strains were grown overnight in LB medium at 37 °C from a single colony for eYFP-T7 RNAP experiments. The overnight cultures were re-inoculated into fresh M9 medium supplemented with 0.4% glycerol, amino acids, and vitamins at 1:100 dilution. The cells were grown for 4.5 h with 0.2% L-rhamnose to induce eYFP-T7 RNAP expression ($OD_{600}$ ~0.2). Then, the cells were pelleted by centrifugation for 1 min and re-suspended in a pre-warmed M9 medium supplemented with 0.4% glucose, amino acids, and vitamins at 1:3 dilution and grown for additional 2 h without L-rhamnose (cell doubling time ~55 min). The change of the medium and the additional growth period without the expression of eYFP-T7 RNAP allows the maturation of the FPs already expressed in cells and the dilution of eYFP-T7 RNAP to a proper concentration in a cell for the single-protein detection. For 6xTetO array measurements, cells were grown at 30 °C in a M9 medium supplemented with 0.4% glucose, amino acids, and vitamins to reach $OD_{600}$ ~0.3. After centrifugation, cells were resuspended to $OD_{600}$ ~0.1 with 0.4% glycerol, amino acids, vitamins, and 0.2% L-arabinose and grown for 3 h to express TetR-eYFP. Then, cells were again pelleted by centrifugation and resuspended to $OD_{600}$ ~0.1 with 0.4% glucose, amino acids, and vitamins and grown for another 3 h (cell doubling time ~99 min). This procedure was required for the maturation of FPs and proper dilution of TetR-eYFP for single-protein imaging. For T7p_*lacZ*-12xTetO experiments, cells were grown overnight in LB medium at 37 °C from a single colony. The overnight cultures were re-inoculated

into a fresh M9 medium supplemented with 0.4% glycerol, amino acids, and vitamins at 1:100 dilution. The cells were grown for 4.5 h with 0.2% L-rhamnose to induce T7 RNAP expression and 0.2% L-arabinose to induce TetR-eYFP expression (OD$_{600}$ ~0.2). Then, the cells were pelleted by centrifugation for 1 min and re-suspended in a pre-warmed M9 medium supplemented with 0.4% glucose, amino acids, and vitamins at 1:3 dilution and grown for additional 2 h without L-arabinose. This additional growth period without the expression of TetR-eYFP allows the maturation of the FPs already expressed in cells and the dilution of TetR-eYFP to a proper concentration in a cell for the TetO array detection.

**Fluorescence microscope and image acquisition**. Cells were pelleted by centrifugation for 1 min and resuspended with a fresh M9 medium supplemented with 0.4% glucose for washing, then pelleted again by centrifugation and resuspended with a fresh M9 medium supplemented with 0.4% glucose in a final volume of 40 μL. Then, 15 μL of the cells was dropped in the center of glass bottom dish (MatTek, P35G-1.0-14-C) coated with a poly-L-lysine (Sigma, P8920) for immobilizing the cells at the surface of the glass, which were incubated for 10 min at room temperature. After removing the non-immobilized cells from the glass surface, 2 mL of a fresh M9 medium supplemented with 0.4% glucose was applied to the glass bottom dish. Samples were imaged on an inverted microscope (Olympus, IX-71) with a 100× oil-immersed objective lens (Olympus) and additional 1.6x amplification. In order to induce transcription, 1 mL of a M9 medium supplemented with 0.4% glucose containing 3 mM IPTG was added to the 2 mL of the medium in the glass bottom dish to make the final concentration of 1 mM IPTG. Phase contrast images and eYFP fluorescence images were acquired at different positions at every 10 s using a cooled EMCCD camera (Andor iXon DU897) equipped with a dichroic mirror (FF520-Di01-25x36, Semrock), an excitation filter (FF03-510/20, Semrock), and an emission filter (HQ550/50m, Chroma). Excitation was provided by an Ar-ion laser (Melles Griot 43 Series Ion Laser) at 514 nm with an intensity of ~0.3 kW cm$^{-2}$. Metamorph software (Molecular Devices) was used to control the automated measurements. All experiments were carried out at room temperature (24 °C) (cell doubling time ~220 min), unless otherwise specified.

As for the measurements using 6xTetO array at 37 °C (cell doubling time ~55 min), 1-mL aliquot of cells was pelleted by centrifugation and resuspended with a fresh M9 medium to be a final volume of 15 μL, and then 1 μL of the cells was placed between a coverslip and a 1.5% low-melting-temperature agarose gel pad (Lonza, #50111) prepared with a M9 medium. The sample chamber was attached to a temperature controller (FCS2, Bioptechs) and maintained at 37 °C during the experiments. To provide a continuous flow of fresh medium, pre-warmed M9 medium supplemented with 0.4% glucose, amino acids, and vitamins was supplied by a syringe pump at 0.3 mL min$^{-1}$ rate. For transcription induction, 10 mM IPTG was added to a flow of a M9 medium supplemented with 0.4% glucose, amino acids, and vitamins and images were acquired after 20 min of IPTG treatment.

For Ksg-treatment experiments, cells were prepared on the glass-bottom dishes. Then, cells were incubated in 2 mL of fresh M9 medium supplemented with 0.4% glucose and 5 mg mL$^{-1}$ Ksg (Sigma-Aldrich, #32354). To induce transcription, 1 mL of a M9 medium supplemented with 0.4% glucose containing 3 mM IPTG and 15 mg mL$^{-1}$ Ksg was added to the 2 mL of the medium in the glass bottom dish to make the final concentration of 1 mM IPTG and 5 mg mL$^{-1}$ Ksg.

**Analysis of fluorescence images**. The images were analyzed using home-built software (Matlab). The program identified the boundaries of individual cells from the phase-contrast images and calculated the total fluorescence intensity of each cell from the fluorescence images automatically. The intensity of auto-fluorescence was obtained from the cells without the plasmid pNL003, which expresses eYFP-T7 RNAP. In the analysis, we excluded cells having too many eYFP-T7 RNAPs, because it is difficult to determine the specific binding above the high fluorescence background. The program found automatically the region where the integrated fluorescence level of 5 × 5 pixels is the maximum in a cell (denoted as ROMIF, Region of Maximum Integrated fluorescence level). The cellular background level of each cell was determined by the average of the fluorescence intensity of whole cellular region except for the ROMIF. Then, the fluorescence intensity of eYFP-T7RNAP, denoted as $\overline{I_{max}}$ where the over-bar denotes the mean over the 5 × 5 area, was calculated by following:

$$\overline{I_{max}} = \overline{\text{Intensity of the ROMIF} - \text{cellular background}}. \quad (1)$$

Then, $\overline{I_{max}}$ of each cell was averaged over all cells at the specific time $t$, $\langle \overline{I_{max}(t)} \rangle$. Average over a cell population is indicated by the angled brackets.

The average fluorescence intensity from the transcribing eYFP-T7 RNAPs bound to DNA was obtained from:

$$I_{transcribing}(t) = \langle \overline{I_{max}(t)} \rangle - \langle \overline{I_{max}(0)} \rangle, \quad (2)$$

where $\langle \overline{I_{max}(0)} \rangle$ is the average of $I_{max}$ obtained from the repressed condition, i.e., without IPTG induction (defined as time = 0).

Finally, the average number of transcribing T7 RNAPs was calculated by dividing the $I_{transcribing}(t)$ by the single eYFP intensity $\langle I_{single} \rangle$, obtained in

Supplementary Fig. 1. Detailed process of how $\overline{\langle I_{single} \rangle}$ was obtained is described in the next section.

**Quantification of fluorescence intensity of a single eYFP**. We constructed a strain expressing membrane-bound Tsr-eYFP (NL003)[32] and quantified the fluorescence intensity of single eYFPs under the same imaging condition as the eYFP-T7 RNAP measurements (Supplementary Fig. 1). Fluorescence intensity from single eYFPs was obtained by measuring the integrated intensity of the fluorescence dot in 5 × 5 pixels (Supplementary Fig. 1a, b). The distribution of the integrated fluorescence intensity (denoted as $\langle I_{single} \rangle$) from each eYFP is shown in Supplementary Fig. 1c. We used $\overline{\langle I_{single} \rangle}$, i.e., $\langle I_{single} \rangle / 25$, for calculating the average number of transcribing T7 RNAP in Fig. 1e. The average number of transcribing T7 RNAP (denoted as $\langle RNAP_{transcribing} \rangle$) was obtained by $\frac{I_{transcribing}(t)}{\overline{\langle I_{single} \rangle}}$.

**Localization of diffraction-limited spots in a living cell**. The location of the diffraction limited dot was determined by a least-squares fit of a 2D Gaussian point spread function to each spot (Supplementary Fig. 4a). The cellular areas identified from the phase-contrast images were fitted with a 2D elliptical function to determine the center, the short, and the long axes of the cells automatically. Two perpendicular lines relative to the cellular centroid were chosen as $x$, $y$ coordinates of each cell (Fig. 2b). Then, the normalized relative position of each dot was determined. We mapped the positions of all foci into the first quadrant of the cell based on the cylindrical symmetry of E. coli cell.

**Measuring in vivo elongation rate of RNAP**. A total of 643 μL of cells in a fresh M9 medium was prepared; 43 μL of cells were withdrawn into 150 μL of pre-cooled RNAlater solution (Ambion, AM7020) before adding IPTG (this is the sample for 0 s). Then, 100 μL of 7 mM IPTG in a M9 medium was added into the remaining 600 μL of cells and 50 μL of cells was withdrawn into 150 μL of pre-cooled RNAlater solution at each time point, typically every 10 s. After sampling, cells were cooled on ice for 20 min. A total of 400 μL of pre-cooled M9 medium was added to cells and gently mixed by pipetting (407 μL of M9 medium was added into 0 s sample). Cells were collected by centrifugation at 10,000 rpm, 4 °C for 2 min. Supernatant was discarded carefully and then the pellets were re-suspended in 100 μL of pre-cooled lysozyme solution (10 mM Tris-HCl pH 8.0, 0.1 mM EDTA, 10 mg mL$^{-1}$ lysozyme (Sigma, L4919)) and vigorously vortexed for 20 s to break cell walls. Then, 0.5 μL of 10% SDS solution (Sigma, L3771) dissolved in 1 mL of RNase-free water (PureLink RNA Mini Kit, Invitrogen, 12183018 A) was added to cells and vortexed for 10 s. Cells were incubated at room temperature for 5 min. 350 μL of lysis buffer (PureLink RNA Mini Kit) containing 1% beta-mercaptoethanol (Sigma, M3148) was added to cells and vortexed for 10 s. Then, total RNA was extracted following the procedures described in the quick reference of PureLink RNA Mini Kit. Using the extracted total RNA, we performed real-time RT-PCR. First-strand cDNAs were synthesized using Superscript III Reverse Transcriptase (Invitrogen, 18180-044) and RNaseOut Recombinant Ribonuclease Inhibitor (Invitrogen, 10777-019). The following reaction mixture was incubated at 50 °C for 60 min, and heat inactivated at 70 °C for 15 min: 3 μL of RNase-free water, 2 μL of total RNA, 2 μL of 5× first-strand buffer, 1 μL of 100 μM reverse primers (N rev or C rev), 0.5 μL of dNTP (invitrogen, 18427-013), 0.5 μL of 0.1 M DTT, 0.5 μL of RNaseOut Recombinant RNase Inhibitor (40 U μL$^{-1}$), and 0.5 μL of superscript III reverse transcriptase (200 U μL$^{-1}$). cDNAs were temporally stored at 4 °C for the next step. For quantitative polymerase chain reaction (qPCR), cDNAs were amplified using TaqMan Universal Master Mix II with UNG (Applied Biosystems, 4440038) and Custom TaqMan Gene Expression Assays (Applied Biosystems) on StepOne Real-Time PCR System (Applied Biosystems, 4376357) with the following reaction mixture condition: 3.5 μL of RT-PCR Grade Water (Ambion, AM9935), 5 μL of TaqMan Universal Master Mix II with UNG, 0.5 μL of Custom TaqMan Gene Expression Assays, and 1 μL of cDNA. The run method contained a holding stage with initial heating at 50 °C for 2 min and an initial denaturation step at 95 °C for 10 min, followed by a cycling stage for 40 cycles with a denaturation step at 95 °C for 15 s and an annealing/amplification step at 60 °C for 1 min. The sequences of the primers and probes used for real-time RT-PCR measurement are summarized in Supplementary Table 1.

**mRNA degradation rate measurement**. E. coli strain BW25993 and lacZΔRBS (without RBS for lacZ gene of BW25993) strains were grown to mid-log phase and cooled to 4 °C for 1 h. Cells were concentrated (OD$_{600}$ × volume ~0.8) with 24 °C M9 medium (supplemented with 0.4% glucose, amino acids, and vitamins), then 1 mM IPTG was added for 30 min to express lacZ mRNA. At time zero, rifampicin was added to the final concentration of 300 ng μL$^{-1}$. Six samples were withdrawn with 2-min interval (0, 2, 4, 6, 8, 10 min) to pre-cooled RNAlater solution. Total RNA purification, first strand cDNA synthesis, qPCR analysis (probe 1 was used to determine the abundance of lacZ mRNA) were performed same as previously described. The result is plotted on a log scale to determine the half-life of lacZ mRNA in our experimental conditions (Supplementary Fig. 15b).

**β-galactosidase assay**. Overnight culture of cells was diluted 1:100 into M9 glycerol medium containing 1 mM IPTG and 0.2% L-rhamnose (SIGMA, R3875).

The culture was grown for 6 h at 37 °C. Miller assays are conducted using the yeast β-galactosidase assay kit (Pierce, 75768). A total of 190 μL of Y-PER reagent was added into 10 μL of cells (cells were diluted or concentrated) and incubated at room temperature for 20 min. Then, 190 μL of M9 glycerol medium and 100 μL of 2× assay buffer were added and incubated at 37 °C. The reaction was stopped with 171 μL of stop solution. Cellular debris was removed by centrifugation for 5 min at $10,000 \times g$ and 500 μL of supernatant was collected. β-galactosidase activity was determined using the following formula: $(1000 \times OD_{420})/(t \times V \times OD_{660})$, where $t$ is the reaction time in minutes, and $V$ is the volume of cells used in the assay, milliliters.

**Single-molecule fluorescence in situ hybridization (smFISH).** We followed the protocol previously reported[58]. A total of 50 oligonucleotides labeled with Atto 594 were hybridized to the target *lacZ* mRNA. Cell cultures were fixed and permeabilized using 70% ethanol. Cells were then hybridized with the probe set, washed, and imaged using epifluorescence microscopy. Overnight culture of cells was diluted 1:200 into a M9 glucose medium supplemented with amino acids and vitamins. The cells were grown for 5 h with 1 mM IPTG to induce lacZ expression ($OD_{600}$ ~0.3). For imaging, excitation was provided by a fiber laser at 580 nm (VFL-P-Series, MPB Communications Inc.). Phase contrast images and fluorescence images for Atto 594 were acquired at multiple positions using a cooled EMCCD camera (Andor iXon DU897). A home-built software (Matlab) was used for image analysis. Phase contrast images were used to identify the cell area and shape. We used Spatzcells[58] in the mRNA fluorescence images to quantify the *lacZ* mRNA in individual cells.

**Reporting Summary.** Further information on research design is available in the Nature Research Reporting Summary linked to this article.

## Data availability
The source data underlying Figs. 1e, 2c, d, h, 3b–e, 4b–d, and 5b–f and Supplementary Figs. 1, 2, 5–7, and 9–15 are provided as a Source Data file.

## Code availability
The analysis codes that were used in this study are available from the corresponding author upon request.

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

## Acknowledgements

We thank J. Xiao for providing the plasmid (pACL08) and P. Choi for helpful discussion. This work was supported by the grant from Midcareer Research Program (NRF-2017R1A2B3010309) of the National Research Foundation of Korea.

## Author contributions

S.Y., A.H.G. and N.K.L. designed the experiments; S.Y., S.K. and N.K.L. wrote the manuscript; S.Y. performed the microscopy experiments. S.Y. and S.K. performed the image analyses and constructed the bacterial strains. S.K. performed real-time RT-PCR experiments. D.-K.K., H.J.A. and J.B.S. helped measurements and analysis. H.J.A. performed the simulation. N.K.L. supervised the project.

## Competing interests

The authors declare no competing interests.
