## [Peer Review File · Nature Communications]

Reviewers' comments:

Reviewer #1 (Remarks to the Author):

This is a mixed review. On the one hand, the paper provides many intriguing new data indicating that transcription of a protein gene causes its radial migration outward towards the periphery of the *E. coli* nucleoid. The methods for locating one particular gene among the many being transcribed are clever. The measurements seem to be carried out carefully and with high precision. I came away convinced that under the conditions studied the transcribed genes really do migrate, albeit by small amounts. However, the presentation is often confusing and I have conceptual problems with the choice of experimental conditions. Interpretation of the results also needs refinement.

1. It is well-established that the size and shape of the nucleoids, the radius of the cytoplasm, and the utilization of RNAP between transcription of *rrn* operons versus transcription of protein genes varies tremendously with growth rate. For fast growth, most RNAP is transcribing *rrn* operons; in slow growth, it is mostly protein genes. It is almost certain that the bright, peripherally located RNAP clusters observed by many groups in fast growth involve transcription of *rrn* operons. The evidence for a peripheral distribution of RNAP clusters in slow growth is less compelling, in my opinion.

The paper seems to make little distinction between these two very different regimes. Clusters of RNAP transcribing *rrn* operons lack ribosomes, yet they are strongly peripheral. RNAPs transcribing protein genes likely include bound polysomes which are very massive. If indeed both lie at the nucleoid periphery, the underlying cause may be very different. This should be clarified in the introduction and the discussion.

2. The growth conditions for different experiments in this paper vary widely. Sometimes the medium is M9; sometimes M9 plus glucose; sometimes M9 plus glycerol. In addition, widely differing temperatures are used, ranging from 24° C to 37° C. Doubling times must vary widely. Nowhere can I find the doubling time stated for any of these conditions. It would be straightforward to use phase contrast to measure cell length versus time for each condition to provide an in situ estimate of the growth rate during the actual microscopy experiments.

3. There is no effort to image the nucleoids. How can we say the genes migrate to the nucleoid periphery without any sense of how large the nucleoid is? The larger the nucleoid radius, the further a gene has to move on average to get near the periphery. Could the membrane stain FM4-64 help determine the radius of the cell each condition? Could a non-perturbing DNA stain such as Sytox help to characterize the nucleoid? Such measurements might help us to get a sense of the magnitude of the shift in $\langle x \rangle$ relative to the size of the nucleoids. The changes in apparent mean radial coordinate $\langle x \rangle$ are of the order of 60-80 nm at their largest (T7 case) and only 30-50 nm for *E. coli* RNAP. How does this compare the size of the nucleoid?

4. In my experience, the cell radius is remarkably constant from cell to cell for a given growth condition. For the transverse coordinate x , that obviates the need to use relative coordinates. It would be better to present histograms on an absolute micron scale for each condition.

According to Fig. 2c, almost no T7 spots lie at $x > 0.5$. Is the nucleoid really so narrow? Or are the phase contrast images overestimating the radius of the entire cell? How is the radius 0.35 chosen for the simulated distributions of Fig. 2?

5. Why were the conditions for the T7 experiments not matched with the conditions for the *E. coli* RNAP experiments? That would help us compare the magnitude of gene migration in both cases, with the size and shape of the nucleoid held constant.

6. It seems interesting that the presence of the gene for the membrane protein LacY does not enhance the shift in $\langle x \rangle$. Does that mean transertion is unimportant, at least for that gene at these expression levels? Is it possible to run experiments in which LacY but not LacZ is present?

7. In Fig. 1e, why is the average number of T7 copies transcribing so small? If only a fraction of a T7 is transcribing in each cell, why are the images so bright? Does the average include cells that exhibit no bright spots?

It seems to me that photobleaching events look identical to unbinding events at the end of a transcript. Could the absence of photobleaching within the kinetics model be driving the apparent number of transcribing copies down?

8. The near absence of radial migration in the delta-RBS case correlates with very low transcription yield, an interesting result in its own right. However, the gene may not migrate because little or no mRNA is being made. This confounds the interpretation of the result as arising from the absence of polysomes bound to the nascent message. The result does not prove that the presence of polysomes when RBS is present is what drives radial migration.

9. For the T7 experiments, we need to see an image like that of Fig. 1c for early times after induction and for later times after induction. Is there easily visible evidence of the migration? (And what was the delay time in the existing Fig. 1c? I can't find it.)

10. The meaning of the coordinate y seems to change in different places. On p. 5, y is the relative long axis coordinate. In Fig. 2e, y has become a radial coordinate like x . Very confusing! Microscope convention would have the radial companion to x denoted as the z coordinate (along the objective axis, perpendicular to the plane of the coverslip))

11. Please indent the paragraphs for clarity.

12. At least in paragraphs comparing $\langle \delta x \rangle$ between different cases, it would be good to include the error estimates in the text.

13. What exactly is meant by the nucleoid periphery? The analysis is carried out on the radial coordinate y . The authors mean to say the genes migrate radially.

14. In both fast and slow conditions, most 70S ribosomes (most translation) occurs in the endcaps. Does the gene also migrate outward axially? If not, then the statement that the migration enhances spatial overlap between mRNA and the translating ribosomes applies to only a minority of translation. We believe that most translation occurs on complete transcripts (unbound to DNA).

15. I believe the drug Ksg jams up the mRNA exit channel within the ribosome. This arrests translation at an early stage of elongation, preventing new initiation events. 70S ribosomes already well into the elongation phase can presumably continue until completion, dissociating into 30S and 50S, but there will be a shortage of open initiation sites to recruit the subunits for subsequent rounds of translation. If that is correct, there will be a much higher fraction of 30S and 50S subunits compared with normal cells. The nucleoid almost surely expands! The $\langle \delta x \rangle$ value may be due to radial nucleoid expansion rather than actual gene migration. I don't trust the interpretation of the Ksg results without nucleoid imaging.

16. In the discussion on line 267, the broad distribution of RNAP within the nucleoid does not imply that transcription initiation begins inside. Much of the binding of RNAP to DNA is non-specific.

17. I found the first rationale for transcription driving radial migration rather vague. Changes in DNA structure near a transcribed gene may indeed occur, but they are very local. By what physical

mechanism does this drive migration over 60 nm?

18. I prefer the second mechanism suggesting that the growth of massive polysomes near an actively transcribed gene drives the migration. This is not due to enhanced translational entropy of the polysomes, however. These polysomes are not free to diffuse—they are bound to DNA! A better idea is that the same excluded volume effects that may help drive polysome-nucleoid segregation exert a radially outward force on the growing polysome chains, which drags the gene outward as well.

19. Is there anything interesting to infer about the time scale of the observed T7 migration, roughly 50-100 s (Fig. 2b)? This is comparable to the time to transcribe a typical protein gene (but maybe that is true only for *E. coli* RNAP—T7 is apparently much faster). Is this time limited by the search for the promoter by the 30 T7 copies? Given the small number of transcribing T7 copies per cell, can we infer that only one polysome is required to drive the radial migration? That is interesting if true.

In summary, I cannot support publication of the paper in anything like its present form. There are interesting results here, but their true nature and importance is obscured by the issues I have raised.

Reviewer #2 (Remarks to the Author):

Overall this manuscript by Yang et al. Provides evidence that transcription-translation coupling relocates actively transcribed loci to the nucleoid periphery in *E. coli*. This fundamental process is important to understand and this paper makes significant advances in the subcellular organization of gene expression processes in bacteria. The authors provide compelling evidence that transcription rate moves the DNA loci towards the nucleoid periphery, but the data that translation is the major factor in this movement is less convincing. Alternative interpretations of the role of translation are not ruled out by the data presented. Additionally, the membrane-tethering of RNase E could potentially explain the differences mentioned and the movement of the DNA loci to the nucleoid periphery may only be indirectly linked to translation. I believe with a few experiments that the authors can more strongly show which factor is most important in the movement of the DNA loci to the nucleoid periphery.

1) As T7 RNA-polymerase doesn't couple with ribosomes, why does T7 produce a stronger movement to the nucleoid periphery than the *E. coli* RNA-polymerase? Wouldn't this suggest the rate of transcription is the major driver and not coupling?

2) As T7 RNA-polymerase doesn't couple with ribosomes, I'm confused as to why the kasugamycin experiment was performed with this strain and not the *E. coli* RNA-polymerase strain. Performing this experiment with the *E. coli* RNA-polymerase would likely yield a much stronger affect on the localization to the nucleoid periphery if their interpretation that translation coupling is the most important factor is correct.

3) As RNase E is responsible for mRNA decay is localized to the inner membrane, and mRNA decay would break the transcription/translation coupling, driving an mRNA to the nucleoid periphery would potentially accelerate mRNA decay and reduce coupling. Constructs lacking the RBS also show lower extent at the periphery, but since its known that translation protects mRNA from decay, couldn't this effect be explained RNase E having enhanced cleavage due to the colocalization with the nascent mRNA? Fast mRNA decay has recently been shown for mRNAs for membrane proteins in *E. coli* that depends on this membrane tether for RNase E (Moffitt Elife 2016). Performing this experiment with a strain lacking the inner membrane tether for RNase E

would be able to distinguish between translation and decay. This may be particularly important since the delta RBS constructs show increased mRNA decay rates.

Other comments:

As the delta RBS lowers translation, but doesn't completely abolish it, wouldn't a stronger effect be observed from a start codon deletion?

Change "dots" to foci, as this is more common terminology.

The manuscript would be improved by significant improvement in english grammar.

on line 144 what are the units for 0.07?

Reviewers' comments:

Reviewer #1 (Remarks to the Author):

This is a mixed review. On the one hand, the paper provides many intriguing new data indicating that transcription of a protein gene causes its radial migration outward towards the periphery of the E. coli nucleoid. The methods for locating one particular gene among the many being transcribed are clever. The measurements seem to be carried out carefully and with high precision. I came away convinced that under the conditions studied the transcribed genes really do migrate, albeit by small amounts. However, the presentation is often confusing and I have conceptual problems with the choice of experimental conditions. Interpretation of the results also needs refinement.

1. It is well-established that the size and shape of the nucleoids, the radius of the cytoplasm, and the utilization of RNAP between transcription of *rrn* operons versus transcription of protein genes varies tremendously with growth rate. For fast growth, most RNAP is transcribing *rrn* operons; in slow growth, it is mostly protein genes. It is almost certain that the bright, peripherally located RNAP clusters observed by many groups in fast growth involve transcription of *rrn* operons. The evidence for a peripheral distribution of RNAP clusters in slow growth is less compelling, in my opinion.

The paper seems to make little distinction between these two very different regimes. Clusters of RNAP transcribing *rrn* operons lack ribosomes, yet they are strongly peripheral. RNAPs transcribing protein genes likely include bound polysomes which are very massive. If indeed both lie at the nucleoid periphery, the underlying cause may be very different. This should be clarified in the introduction and the discussion.

Answer: We appreciate Reviewer 1's comments and excellent suggestion. As Reviewer 1 suggested, we revised the main text to clarify the distinction between these two different transcription regimes.

Introduction

At the end of the Introduction section, we clearly mentioned that transcription-translation coupling is the major factor responsible for protein gene movement (page 3, highlighted in yellow color).

“Three factors are involved in gene locus movement during transcription and responsible for the relocation of gene loci during transcription to the nucleoid periphery: transcriptional activity,

ribosome binding to mRNAs, and transcription-translation coupling. Among these factors, transcription-translation coupling is the major factor determining **protein** gene locus movement in *E. coli* RNAP transcription.”

Discussion

In the Discussion section, we clearly distinguish between the mechanisms of *rrn* operon and protein gene transcription (page 12, highlighted in yellow color).

“Jin *et al.* observed foci of transcribing RNAP located at the surface of the nucleoid under fast growth conditions.^{20,29} These foci were suggested to be the locations of actively transcribing genes, such as *rrn* operons. Since ribosomal RNAs of *rrn* operons are not translated by ribosomes, RNAP foci under fast growth conditions may occur due to high transcriptional activity without transcription-translation coupling. A recent live-cell super-resolution imaging study showed that highly transcribing RNAPs tend to cluster at the edge of the nucleoid in minimal media growth conditions.²³ In this case, the gene locus may have moved to the nucleoid periphery mainly by transcription-translation coupling with forming a large DNA-RNAP-mRNA-ribosome complex.”

2. The growth conditions for different experiments in this paper vary widely. Sometimes the medium is M9; sometimes M9 plus glucose; sometimes M9 plus glycerol. In addition, widely differing temperatures are used, ranging from 24° C to 37° C. Doubling times must vary widely. Nowhere can I find the doubling time stated for any of these conditions. It would be straightforward to use phase contrast to measure cell length versus time for each condition to provide an in situ estimate of the growth rate during the actual microscopy experiments.

Answer: We thank Reviewer 1 for these comments. As Reviewer 1 noted, we found that we didn't clearly describe the reason for the specific growth conditions used in the original manuscript. In this revised manuscript, we described the growth conditions clearly and reported the cell doubling time, which have been added to the Methods section.

For the expression of eYFP-T7 RNAP and TetR-eYFP by L-rhamnose and L-arabinose, respectively, we used M9 medium supplemented with 0.4% glycerol, amino acids, and vitamins. Then, the cells were grown in M9 medium supplemented with 0.4% glucose, amino acids, and vitamins for 2 or 3 hours (cell doubling time ~ 55 min) before microscopic measurements. M9 medium with glucose was used for two reasons. First, the expression levels of eYFP-T7 RNAP and TetR-eYFP were too high for

single-protein imaging. We therefore used M9 medium supplemented with 0.4% glucose, amino acids, and vitamins without inducers, which efficiently blocked the expression of eYFP-T7 RNAP and TetR-eYFP compared to that in cells grown in M9 medium supplemented with glycerol. Then, the cells were grown for 2 hours or 3 hours to allow for the maturation for eYFP and to dilute the concentrations of eYFP-T7 RNAP and TetR-eYFP for single-protein imaging as specified in the original manuscript. We have revised the manuscript to clarify this point in the Method section.

For microscopic measurements using a gel pad at 37 degrees, we used M9 medium supplemented with 0.4% glucose, amino acids, and vitamins, not M9 medium. We thank Reviewer 1 for pointing out this mistake, which has been corrected in the revised manuscript. Under this condition, we measured the doubling time using phase contrast images and found it to be 55 minutes. The doubling time measured in the culture grown while shaking was also 55 minutes. We have revised the manuscript and added the doubling time.

For microscopic measurements using a poly-L-lysine coated glass bottom dish at 24 degrees, we used M9 medium supplemented with 0.4% glucose, not M9 medium. We have corrected this mistake in the revised manuscript. In this case, the use of M9 medium supplemented with 0.4% glucose gave a lower background signal than M9 medium supplemented with 0.4% glucose, amino acids, and vitamins. The experiments performed on a poly-L-lysine coated glass bottom dish were conducted for approximately 10 min of observation in M9 medium supplemented with 0.4 % glucose. The cells started to become detached from the glass surface after approximately 20 minutes. Thus, the doubling time on the actual microscope could not be measured at 24 degrees. We measured the doubling time of the cells grown while shaking in the same culture medium and at the same temperature, which was approximately 220 min. We have revised the manuscript and added the doubling time.

The revised parts are marked by yellow highlight in “Cell growth conditions” and “Fluorescence microscope and image acquisition” sections in the revised manuscript (page 19 & 20, highlighted in yellow color).

3. There is no effort to image the nucleoids. How can we say the genes migrate to the nucleoid periphery without any sense of how large the nucleoid is? The larger the nucleoid radius, the further a gene has to move on average to get near the periphery. Could the membrane stain FM4-64 help determine the radius of the cell each condition? Could a non-perturbing DNA stain such as Sytox help to characterize the nucleoid? Such measurements might help us to get a sense of the magnitude of the

shift in $\langle x \rangle$ relative to the size of the nucleoids. The changes in apparent mean radial coordinate $\langle x \rangle$ are of the order of 60-80 nm at their largest (T7 case) and only 30-50 nm for *E. coli* RNAP. How does this compare the size of the nucleoid?

Answer: We thank Reviewer 1 for the excellent suggestion. As Reviewer 1 suggested, we imaged the nucleoid using SYTOX-Green dye and quantified the nucleoid size to determine the magnitude of gene movement in the x-axis relative to the size of nucleoid.

Fig. P1. Nucleoid imaging using SYTOX-Green dye. (a) A representative cell shown in a phase contrast overlaid with SYTOX fluorescent image (Scale bar, 500 nm). The nucleoid width and length are defined as the full width at half maximum of the intensity profile along the short and long axes of the cell, respectively. (b-c) Comparison of the nucleoid (b) width and (c) length in each experimental condition. (For each box, center spot, mean; center line, median; box limits, 25th and 75th percentiles; whiskers, 1.5x interquartile range)

We imaged the nucleoid using SYTOX-Green dye, which is used for the non-perturbative imaging of chromosomal DNA in live *E. coli* cells (Bakshi et al, *Appl. Environ. Microbiol.* **80**, 4977-4986, 2014). SYTOX-Green dye (Invitrogen #S7020) was added to the cell culture at a final concentration of 500 nM and the cells were grown by shaking for 1 hour. Before imaging, the cells were washed twice to remove free dye. The nucleoid width and length were defined as the full width at half maximum of the intensity profiles of SYTOX fluorescent images along the short and long axes of the cell, respectively, as described by Bakshi et al. (Bakshi et al, *Appl. Environ. Microbiol.* **80**, 4977-4986, 2014) (Fig. P1a). As a result, we confirmed that the size of the nucleoids was maintained for the various measurements during both T7 RNAP- and *E. coli* RNAP-mediated transcription (width = $0.60 \pm 0.01 \mu\text{m}$ and length = $1.53 \pm 0.04 \mu\text{m}$) in our experimental conditions.

We measured the cell size using phase-contrast images as previously described (Bakshi et al, *Mol. Microbiol.* **94**, 871-8, 2014; Sanamrad et al, *Proc. Natl. Acad. Sci. U S A* **111**, 11413-8, 2014; Stracy et al, *Proc. Natl. Acad. Sci. U S A* **112**, E4390-E4399, 2015). The cell size on the x-axis (short length) was $1.09 \pm 0.10 \mu\text{m}$. Thus, the length of the cell from the center is $0.55 \mu\text{m}$, and the size of the nucleoid from the cell center is $0.30 \mu\text{m}$. As a result, the movements of genes during transcription with *E. coli* RNAP (30-50 nm) and T7 RNAP (60-80 nm) corresponded to 10%-16% and 20%-27% of the nucleoid size, respectively.

We have added these new experimental results of Fig. P1 in the revised manuscript (Supplementary Fig. 12). In addition, we added the following sentences to the main text (page 9, highlighted in yellow color).

“Then, we measured the extent of gene locus movement relative to the size of the nucleoid. We imaged the nucleoid using SYTOX-Green dye⁴³. The nucleoid width and length were defined as the full width at half maximum of the intensity profiles of SYTOX fluorescent images, as described by Bakshi et al.⁴³ (Supplementary Fig. 12). The size of the nucleoids was maintained for the various measurements using both T7 RNAP and *E. coli* RNAP (short axis = $0.60 \pm 0.01 \mu\text{m}$). Thus, the width of the nucleoid from the center of the cell was 300 nm. As a result, gene movements during *E. coli* RNAP transcription (30-50 nm) and T7 RNAP transcription (60-80 nm) correspond to 10-16 % and 20-27 % of the nucleoid size (width), respectively.”

4. In my experience, the cell radius is remarkably constant from cell to cell for a given growth condition. For the transverse coordinate x , that obviates the need to use relative coordinates. It would be better to present histograms on an absolute micron scale for each condition.

Answer: We thank Reviewer 1 for this comment. As Reviewer 1 noted, the cell radii were almost constant in the cell population. However, the observed cell radii showed a distribution with a standard deviation. In our case, the ratio of the standard deviation (s.d.) to the mean was approximately 9% ($547 \pm 48 \text{ nm}$ for cells in Fig. 2c). Figure P2 shows a comparison of gene locus movement measured with the relative and absolute cell radii. There was a 9% difference in the distance moved by the gene

locus. Therefore, we would like to normalize the cell radius to obtain more accurate subcellular DNA locus positions. For this reason, other studies have also used relative cell sizes (Sanamrad et al, *Proc. Natl. Acad. Sci. U S A* **111**, 11413-8, 2014; Stracy et al, *Proc. Natl. Acad. Sci. U S A* **112**, E4390-E4399, 2015; Wang et al, *Science* **333**, 1445-9, 2011) .

Fig. P2. Comparison of gene locus movement measured as a relative value and an absolute value in the T7p_lacZ-12xTetO strain (Fig. 4d). The movement was measured to be 81 ± 14 nm as a relative value and 74 ± 13 nm as an absolute value.

According to Fig. 2c, almost no T7 spots lie at $x > 0.5$. Is the nucleoid really so narrow? Or are the phase contrast images overestimating the radius of the entire cell? How is the radius 0.35 chosen for the simulated distributions of Fig. 2?

Answer: We thank Reviewer 1 for this comment. As Reviewer 1 suggested, we measured both the nucleoid radius and the cell radius. We found that the ratio of the nucleoid radius to the cell radius was ~ 0.55 ($0.60 \mu\text{m} / 1.09 \mu\text{m}$) in our experimental conditions. This value seems to be consistent with the distributions shown in Fig. 2c and 2d.

In the simulation, a radius of 0.35 gave the distribution closest to the experimental distribution. We revised the manuscript to clearly show this in the main text (page 6, highlighted in yellow color).

“we randomly generated 100,000 spots with boundary conditions of $-l < x_i, y_i < l$ and a radial distance ($r = \sqrt{x_i^2 + y_i^2} < l$) to mimic nucleoid condensation and gave a 70-nm localization error for each spot (Fig. 2e). **When l equaled 0.35**, the simulated distribution in the x-axis was very similar to that in Fig.

2c, which shows the initial transcription distribution (Fig. 2f).”

5. Why were the conditions for the T7 experiments not matched with the conditions for the *E. coli* RNAP experiments? That would help us compare the magnitude of gene migration in both cases, with the size and shape of the nucleoid held constant.

Answer: We thank Reviewer 1 for this comment. The T7 RNAP experimental conditions did not match those of the *E. coli* RNAP experiments because each experiment required different growth conditions for single-protein imaging. The concentration of fluorescent protein (FP) labeled T7 RNAP in the cell was very important to detect a transcribing T7 RNAP as a diffraction limited spot in a cell (Fig. 1, 2, and 4f). Thus, we had to optimize the growth condition carefully to maintain the proper level of T7 RNAP expression, as described in detail in the answer to question 2. For the *E. coli* RNAP experiments, we used a TetO array with FP labeled TetR that was obtained from the Xiao group (Johns Hopkins University). In the same manner, an appropriate level of FP-TetR expression was important to detect the DNA loci clearly. Xiao and coworkers have reported experimental conditions to detect DNA loci using FP-TetR (Hensel et al, *PLoS Biol.* **11**, e1001591, 2013). Thus, we followed their protocol. As a result, we could not use the same growth conditions for T7 RNAP transcription and *E. coli* RNAP transcription. However, we always used M9 minimal medium for cell growth. As Reviewer 1 suggested, we have reported the cell doubling time in the revised manuscript, which may guide comparisons of the results to some degree. By measuring the size of the nucleoid, as shown in Fig. P1, we also showed that the nucleoid radii were not different between T7 RNAP- and *E. coli* RNAP-driven transcription.

6. It seems interesting that the presence of the gene for the membrane protein LacY does not enhance the shift in $\langle x \rangle$. Does that mean transertion is unimportant, at least for that gene at these expression levels? Is it possible to run experiments in which LacY but not LacZ is present?

Answer: We appreciate Reviewer 1’s comments and suggestion. As Reviewer 1 noted, Figure 4f shows that the transertion of LacY did not change the distance moved by the gene locus. Therefore, transertion is not a major factor in gene locus movement, at least in our experimental conditions. However, these data are not direct evidence that transertion is not important for gene locus movement overall.

As Reviewer 1 suggested, we prepared a strain with the *lacY* gene only (T7p_ *lacY*-12xTetO). In this

strain, LacY is expressed by T7 RNAP, and the location of the gene locus was detected by using eYFP-TetR. As shown in Fig. P3, the gene locus movement was also observed for the strain expressing only LacY. However, the size of LacY gene locus movement was comparable to that of LacZ gene locus movement. This result again shows that LacY gene locus also moves by the transcription.

Fig. P3. Average relative x-positions of gene loci in the T7p_lacZ-12xTetO and T7p_lacY-12xTetO strains.

To understand the effect of transterion on gene locus movement, additional experiments in various experimental conditions are required.

7. In Fig. 1e, why is the average number of T7 copies transcribing so small? If only a fraction of a T7 is transcribing in each cell, why are the images so bright? Does the average include cells that exhibit no bright spots?

Answer: We thank Reviewer 1 for this comment. As reviewer 1 noted, not all cells exhibit bright spots indicating transcription. We included cells that exhibited no bright spots in the average number of transcribing T7 RNAPs in Fig. 1e, as described in the Methods section, because we defined this parameter as the average number of transcribing T7 RNAPs per cell. As shown in Fig. 1c, the signal-to-noise ratio of a single eYFP-T7 RNAP is close to 4, which typically produces an image with good contrast. We also chose the image area to show many clearly visible diffraction-limited spots

indicating transcription in the cells in Fig. 1e. Thus, in the revised manuscript, we added a more detailed explanation to the caption of Fig. 1e as follows:

“Average number of transcribing eYFP-T7 RNAPs during transcription per cell after IPTG induction, including cells that exhibit no bright spots.”

It seems to me that photobleaching events look identical to unbinding events at the end of a transcript. Could the absence of photobleaching within the kinetics model be driving the apparent number of transcribing copies down?

Answer: We thank Reviewer 1 for this comment. As Reviewer 1 noted, the apparent number of transcribing RNAPs may be underestimated due to photobleaching. We imaged different cells at each time point to avoid issues due to photobleaching as much as possible.

Thus, we did not use the average number of transcribing RNAPs for data interpretation. We used the data in Fig. 1e to measure the elongation rate of T7 RNAP, which is nearly independent of the average number of transcribing T7 RNAPs. We also confirmed that the elongation rate shown in Fig. 1e matched well with the elongation rate measured by RT-PCR (Supplementary Fig. 2), as described in the main text.

Because Reviewer 1 commented on this issue, we added the following sentence to the main text (page 5, highlighted in yellow color) of the revised manuscript to indicate the potential effect of photobleaching on the average number of transcribing T7 RNAPs.

“It is possible that the average number of transcribing T7 RNAPs in Fig. 1e was underestimated due to the photobleaching effect.”

8. The near absence of radial migration in the delta-RBS case correlates with very low transcription yield, an interesting result in its own right. However, the gene may not migrate because little or no mRNA is being made. This confounds the interpretation of the result as arising from the absence of polysomes bound to the nascent message. The result does not prove that the presence of polysomes when RBS is present is what drives radial migration.

Answer: We thank Reviewer 1 for this comment. Regarding *E. coli* RNAP transcription, we conclude that the formation of “polysomes” of a DNA-RNAP-mRNA-ribosome complex is not the driving force of gene locus movement, as the Reviewer noted. The results in the RBS-deleted strains showed that transcription-translation coupling enhances the transcriptional activity significantly (16-24 fold),

which is the main reason for the 2.6-fold increase in gene locus movement compared with that observed in the RBS-deleted strains (no transcription-translation coupling).

The effect of “polysome” formation on gene locus movement is shown in Fig. 4e and 4f. Here, we used T7 RNAP transcription. T7 RNAP has no physical interaction with ribosomes and moves much faster than ribosomes. When we deleted the RBS, gene loci moved 35% less than those in the strain with the RBS. Thus, ribosome binding to mRNA enhances gene locus movement by only 30-35% of the total distance.

Thus, our final conclusion on *E. coli* RNAP transcription was the following: **“Thus, transcription-translation coupling increased transcriptional activity relative to the decoupled case (without the RBS). The enhanced transcriptional activity with transcription-translation coupling may have increased gene locus movement by more than 2.6-fold.”**

These results and a discussion are described in the main text in the section “Analysis of the factors contributing to gene locus movement by transcription-translation coupling” (page 11, highlighted in yellow color).

9. For the T7 experiments, we need to see an image like that of Fig. 1c for early times after induction and for later times after induction. Is there easily visible evidence of the migration? (And what was the delay time in the existing Fig. 1c? I can’t find it.)

Answer: We thank Reviewer 1 for the comment. As Reviewer 1 suggested, the diffraction-limited spots were more often localized close to the center of the cell in images obtained at early times (< 50 sec after induction). As expected, however, the number of observed spots was very low. The images obtained late after induction (> 270 sec) clearly showed that the spots were more often localized close to the cell membrane. As reviewer 1 suggested, we modified Supplementary Fig. 3 to provide visible evidence of migration as follows:

Supplementary Figure 3. Additional representative images of T7p_4.5kb cells after adding IPTG. When the image was obtained relative to transcription induction by IPTG is specified in each image. The images obtained at early times showed diffraction-limited spots localized close to the center of the cells, but few spots were observed. After 200 sec of IPTG induction, diffraction-limited spots localized close to the plasma membrane were observed more frequently. Scale bar, 2 μm .

We also specified the induction time in the caption of Fig. 1c. **“The image was acquired at 303 sec after adding IPTG.”** Furthermore, we specified the induction time in the caption of Fig. 2a. **“Representative images of T7p_4.5kb cells acquired late (> 270 sec) after adding IPTG.”**

10. The meaning of the coordinate y seems to change in different places. On p. 5, y is the relative long axis coordinate. In Fig. 2e, y has become a radial coordinate like x. Very confusing! Microscope convention would have the radial companion to x denoted as the z coordinate (along the objective axis, perpendicular to the plane of the coverslip))

Answer: We thank Reviewer 1 for this comment. As Reviewer 1 noted, Figure 2e was confusing. Thus, we modified Fig. 2e according to Reviewer 1’s suggestion.

11. Please indent the paragraphs for clarity.

Answer: We have revised the manuscript as Reviewer 1 suggested.

12. At least in paragraphs comparing between different cases, it would be good to include the error estimates in the text.

Answer: We thank Reviewer 1 for this suggestion. As Reviewer 1 suggested, we have included the errors (mostly standard deviation) in the text.

“Interestingly, relocation of the gene locus under transcription was dramatically reduced in the RBS-deleted strain (Fig. 3b, gray); the movement of the *lacZ* gene locus with the RBS was approximately 32 ± 9 nm, while nearly no movement was observed for the *lacZ* gene locus without the RBS (2 ± 4 nm) (Fig. 3d). Then, we enhanced the gene expression level by increasing the temperature to 37 degrees (Fig. 3e). As the expression level increased, the distance moved by the gene locus with the RBS (*lacZ*-6xTetO) increased to 48 ± 7 nm (Fig. 3e, blue bars). Again, when the RBS was deleted (*lacZ*-6xTetO_ΔRBS), the distance moved by the gene locus was dramatically reduced to 18 ± 11 nm (Fig. 3e, gray bar).”

“Indeed, the distance moved by the *LacZ* gene locus was significantly increased to 81 ± 14 nm at 24 degrees (Fig. 4b-d) relative to the 32 ± 9 nm distance driven by *E. coli* RNAP, as shown in Fig. 3d.”

“The blocking of translation initiation resulted in a reduced gene locus movement of 59 ± 2 nm (Fig. 4e and Supplementary Fig. 13), which is approximately 30 % less than the movement observed without Ksg (Fig. 4d).”

“The half-life of *LacZ* mRNA was 5.5 ± 0.8 min, which was reduced to 2.4 ± 0.3 min for the RBS-deleted strain (Supplementary Fig. 15b).”

13. What exactly is meant by the nucleoid periphery? The analysis is carried out on the radial coordinate y . The authors mean to say the genes migrate radially.

Answer: We thank Reviewer 1 for this comment. As Reviewer 1 noted, the nucleoid periphery in our study means mostly the radial edge. Previous studies have used the expression “nucleoid periphery” (Cabrera & Jin, *Mol. Microbiol.* **50**, 1493-505, 2003; Stracy et al, *Proc. Natl. Acad. Sci. U S A* **112**, E4390-E4399, 2015). Thus, we decided to use “nucleoid periphery” for consistency with other studies.

We clearly mention in the revised manuscript that movement to the nucleoid periphery means radial movement (page 6, highlighted in yellow color).

“Interestingly, transcription foci that appeared between 300 sec and 350 sec after induction moved to the nucleoid periphery (**radial movement**), and their distribution was significantly different compared with their initial distribution (Fig. 2d).”

14. In both fast and slow conditions, most 70S ribosomes (most translation) occurs in the endcaps. Does the gene also migrate outward axially? If not, then the statement that the migration enhances spatial overlap between mRNA and the translating ribosomes applies to only a minority of translation. We believe that most translation occurs on complete transcripts (unbound to DNA).

Answer:

We thank Reviewer 1 for this comment. In high-level expression conditions, the gene locus also moves axially. Figure P4 shows the distribution of the *lacZ* locus in the axial direction (long length), measured in the T7p_ *lacZ*-12×TetO strain (Fig. 4a-c). The *lacZ* locus moved outward by 78 ± 32 nm (Fig. P4). However, no significant gene locus movement in the axial direction was observed in other experimental conditions in which less mRNA was expressed compared with that in the T7p_ *lacZ*-12×TetO strain. Thus, movement of the gene locus increases the spatial overlap between mRNA and the ribosome.

Fig. P4. Distributions of the locations of the *lacZ* gene locus along the axial coordinate (longer length of *E. coli* cell) for T7p_ *lacZ*-12xTetO strain in Fig. 4a. The gray line denotes the distribution of the gene locus with a basal level of T7 RNAP and no IPTG. The transcription by T7 RNAP induced the movement of the gene locus outwardly in the axial direction by 78 ± 32 nm.

We have added Fig. P4 as Supplementary Fig. 11 into the revised manuscript. We also added the following sentences to the main text (page 9, highlighted in yellow color).

“In addition, the outward movement of the *lacZ* gene locus in the axial direction (78 ± 32 nm) was also observed in the T7p_ *lacZ*-12xTetO strain (Supplementary Fig. 11), which was not observed in other experimental conditions that expressed less amounts of mRNAs compared with T7p_ *lacZ*-12xTetO strain.”

15. I believe the drug Ksg jams up the mRNA exit channel within the ribosome. This arrests translation at an early stage of elongation, preventing new initiation events. 70S ribosomes already well into the elongation phase can presumably continue until completion, dissociating into 30S and 50S, but there will be a shortage of open initiation sites to recruit the subunits for subsequent rounds of translation. If that is correct, there will be a much higher fraction of 30S and 50S subunits compared with normal cells. The nucleoid almost surely expands! The value may be due to radial nucleoid expansion rather than actual gene migration. I don't trust the interpretation of the Ksg results without nucleoid imaging.

Answer: We appreciate this excellent suggestion by Reviewer 1. According to Reviewer 1's suggestion, we compared the nucleoid morphologies before and after kasugamycin (KSG) treatment using SYTOX Green for nucleoid imaging. KSG treatment did not lead to nucleoid expansion (Fig. P5) in our experimental conditions. Thus, we observed (Fig. 4e) actual gene locus movement. In a

study by Bakshi et al. (Bakshi et al, *Mol. Microbiol.* **94**, 871-87, 2014), 5 mg/ml Ksg treatment induced a change in nucleoid width of approximately 5% at 30 degrees (cell doubling time ~ 48 min). We also used 5 mg/ml Ksg, but cells were grown at 24 degrees in minimal medium (cell doubling time ~ 220 min). The different experimental conditions might have caused different changes to nucleoid morphology.

We would like to emphasize that the purpose of the experiment using Ksg was to assess the effect of ribosome binding to mRNA on gene locus movement (Fig. 4e). Here, we found that the blocking of ribosome binding to mRNA resulted in 30% less movement of gene loci. Because antibiotics, like Ksg, often have global effects on the cell, we also used RBS-deleted mRNA to block the binding of ribosomes to mRNA (Fig. 4f). The RBS-deleted strains showed 35% less gene locus movement. Thus, we tested the ribosome binding effect of not only Ksg but also the use of RBS-deleted strains. Because the results in both conditions were similar, we confirmed that the blocking of ribosome binding decreases gene locus movement by 30-35% during T7 RNAP transcription.

Fig. P5. Comparison of the nucleoid (a) width and (b) length with and without 5 mg/ml Ksg treatment (after 15 min of incubation). (For each box, center spot, mean; center line, median; box limits, 25th and 75th percentiles; whiskers, 1.5x interquartile range)

This new result from an experiment suggested by Reviewer 1 also supports our conclusion on gene movement; we have added the results in Fig. P5 to the revised manuscript (Supplementary Fig. 14). In addition, we added the following sentences to the main text (page 10, highlighted in yellow color).

“The blocking of translation initiation resulted in a reduced gene locus movement of 59 ± 2 nm (Fig. 4e and Supplementary Fig. 13), which is approximately 30 % less than the movement observed without Ksg (Fig. 4d). Since it has been known that Ksg treatment leads to the change in the nucleoid morphology⁴⁴, we assessed whether the nucleoid morphology before and after Ksg

treatment differ and observed no change in the nucleoid morphology in our experimental condition (Supplementary Fig. 14).”

16. In the discussion on line 267, the broad distribution of RNAP within the nucleoid does not imply that transcription initiation begins inside. Much of the binding of RNAP to DNA is non-specific.

Answer: We thank Reviewer 1 for this comment. We have modified the sentence as follows (page 12, highlighted in yellow color):

“Transcription initiation via the binding of RNAP to a promoter occurs in the nucleoid region. A recent single-particle tracking experiment with ribosomes demonstrated that free ribosomal subunits are not excluded from the *E. coli* nucleoid.”

17. I found the first rationale for transcription driving radial migration rather vague. Changes in DNA structure near a transcribed gene may indeed occur, but they are very local. By what physical mechanism does this drive migration over 60 nm?

Answer: We thank Reviewer 1 for this comment. As Reviewer 1 noted, the physical mechanism by which genes move following a conformational change in DNA is unknown, and changes in chromosomal DNA conformation in transcription may be local. Thus, we moved the text regarding the entropic force and excluded volume effect to the first part of the Discussion section and suggested DNA structural change as a second possible source of gene movement.

However, we cannot exclude effect of DNA conformational change on gene locus movement. For example, transcribing RNAPs generate 1-10 supercoils/second (Lavelle, *Curr. Opin. Genet. Dev.* **25**, 74-84, 2014), and the change in end-to-end DNA extension is close to 56 nm per superhelical turn (Revyakin et al, *Proc. Natl. Acad. Sci. U S A* **101**, 4776-4780, 2004). Thus, DNA supercoiling generated by transcribing RNAPs can induce structural changes in DNA that range in size from tens of nanometers to hundreds of nanometers. In addition, DNA supercoiling is highly dynamic structure that can move by diffusion and hopping process (van Loenhout et al, *Science* **338**, 94-7, 2012) on a millisecond time scale, affecting several kilobases away from their origin (Moulin et al, *Mol. Microbiol.* **55**, 601-610, 2005). Thus, it is possible that transcription-induced DNA supercoiling may induce structural changes in chromosomal DNA that are broader than we thought. In addition, recent models of the *E. coli* chromosome have been developed at nucleotide resolution within a volume

corresponding to that of a cell; the extensive protein occupancy domains that are enriched in RNAPs are suggested to be shifted toward the nucleoid periphery (Hacker et al, *Nucleic Acids Res.* **45**, 11043-11055, 2017). This result is also consistent with the experimental data (Stracy et al, *Proc. Natl. Acad. Sci. U S A* **112**, E4390-E4399, 2015). Thus, we cannot exclude DNA supercoiling generated by transcription as a possible source of DNA movement toward the nucleoid periphery.

We modified the Discussion section in the main text as follows (page 13, highlighted in yellow color):

“Supercoiling due to high RNAP activity would accumulate very rapidly before timely correction by topoisomerases and gyrase, which may change the local DNA structure size range from tens of nanometers to hundreds of nanometers^{53,54}. In addition, DNA supercoiling is highly dynamic that can move by diffusion and hopping process⁵⁵, affecting several kilobases away from the origin of the supercoiling⁵⁶.”

18. I prefer the second mechanism suggesting that the growth of massive polysomes near an actively transcribed gene drives the migration. This is not due to enhanced translational entropy of the polysomes, however. These polysomes are not free to diffuse—they are bound to DNA! A better idea is that the same excluded volume effects that may help drive polysome-nucleoid segregation exert a radially outward force on the growing polysome chains, which drags the gene outward as well.

Answer: We thank Reviewer 1 for this helpful comment. As Reviewer 2 suggested, we modified the text in the Discussion section to clarify the mechanism.

“In this model, DNA polymers are confined in the center of the cell by conformational entropy, while multiple 70S-polysomes formed on nascent mRNA inside the nucleoid diffuse outside the nucleoid due to the excluded volume effect.”

We also moved the text regarding the entropic force and excluded volume effect to the earlier part of the revised manuscript and then suggested DNA structural changes as a second possible source of gene movement (page 12, highlighted in yellow color).

19. Is there anything interesting to infer about the time scale of the observed T7 migration, roughly 50-100 s (Fig. 2b)? This is comparable to the time to transcribe a typical protein gene (but maybe that

is true only for *E. coli* RNAP—T7 is apparently much faster). Is this time limited by the search for the promoter by the 30 T7 copies? Given the small number of transcribing T7 copies per cell, can we infer that only one polysome is required to drive the radial migration? That is interesting if true.

Answer: We appreciate Reviewer 1's comments. The time scale shown in Fig. 2b represents the elongation rate. It takes approximately 54 sec for T7 RNAP to move from the promoter to the termination site. This is fully explained in Fig. 1e. The time scale we believe Reviewer 1 is interested in is shown in Fig. 2h and refers to the migration time of the gene locus in transcription. It takes approximately 200 sec for the gene locus to undergo its full movement to the edge of the nucleoid. It would be very interesting to determine the meaning of this time scale, but it is currently difficult to understand the biological meaning of this time scale. We will investigate this in our future work.

Regarding the question about the polysome, our data indicate that only one polysome (DNA-RNAP-mRNA-ribosome complex) is required to drive gene locus movement. The average time for a single T7 RNAP among 35 copies of T7 RNAP to adopt an elongation state was approximately 131 sec, which was longer than the elongation time of T7 RNAP (~ 50 sec) in our imaging conditions (Fig. 1e). Thus, the detected diffraction-limited spots in our eYFP-T7 RNAP system (Fig. 1, 2, and 4f) mostly represent a single T7 RNAP transcribing the *lacZ* gene. The formation of only one polysome seems to be required to drive the radial migration of the gene locus, as Reviewer 1 noted. We have added the following sentence to the revised manuscript:

(Page 10, highlighted in yellow color)

“Considering that the spots we observed contain mostly single T7 RNAP (Fig. 1e), one DNA-RNAP-mRNA-ribosome complex seems to be required to drive movement of the transcribing gene locus toward the nucleoid periphery.”

Reviewer #2 (Remarks to the Author):

Overall this manuscript by Yang et al. Provides evidence that transcription-translation coupling relocates actively transcribed loci to the nucleoid periphery in *E. coli*. This fundamental process is important to understand and this paper makes significant advances in the subcellular organization of gene expression processes in bacteria. The authors provide compelling evidence that transcription rate moves the DNA loci towards the nucleoid periphery, but the data that translation is the major factor in this movement is less convincing. Alternative interpretations of the role of translation are not ruled out

by the data presented. Additionally, the membrane-tethering of RNase E could potentially explain the differences mentioned and the movement of the DNA loci to the nucleoid periphery may only be indirectly linked to translation. I believe with a few experiments that the authors can more strongly show which factor is most important in the movement of the DNA loci to the nucleoid periphery.

1) As T7 RNA-polymerase doesn't couple with ribosomes, why does T7 produce a stronger movement to the nucleoid periphery than the *E. coli* RNA-polymerase? Wouldn't this suggest the rate of transcription is the major driver and not coupling?

Answer: We thank Reviewer 2 for this comment. T7 RNAP induces more movement than *E. coli* RNAP because of the high transcriptional activity of T7 RNAP. In T7 RNAP transcription, as Reviewer 2 commented, the transcription rate is the major factor causing the relocation of gene loci, and the magnitude of the movement depends on transcriptional activity (Fig. 4b-d). We showed that ribosome binding to mRNA generated by T7 RNAP increases gene locus movement by only 35-35% (Fig. 4f) for T7 RNAP transcription using the ribosome binding site (RBS)-deleted strains. This result indicates that without binding of the ribosome to mRNA, the gene locus moved to the nucleoid periphery. Thus, transcriptional activity is the major driving force of T7 RNAP transcription, as noted by Reviewer 2.

However, *E. coli* RNAP transcription is different than T7 RNAP transcription. Transcription-translation coupling is the major factor driving gene locus movement in *E. coli* RNAP transcription. This is clearly shown in Fig. 3d and 3e; when transcription-translation coupling was disrupted by deletion of the ribosome binding site (RBS), the gene locus moved only 18 nm at 37 degrees. With the RBS, gene locus movement was increased to 48 nm. Thus, transcription-translation coupling increases gene locus movement by 2.6-fold. In addition, as Reviewer 2 suggested, we prepared strains with no RBS and no start codon (Fig. P8), which exhibited a stronger translation inhibition effect compared with that in the RBS-deleted strain (Fig. P8b). In this case, nearly no movement of gene loci was observed. Thus, transcription-translation coupling is important for *E. coli* RNAP transcription.

Because *E. coli* RNAP transcription is often regarded as more important than T7 RNAP transcription as a model system of bacterial transcription, we explained and discussed gene locus movement mostly based on the results of *E. coli* RNAP transcription. However, we believe that clearly mentioning that transcription is the major driving force for T7 RNAP transcription will help potential readers better understand our results.

Thus, we added the following sentence (page 10, highlighted in yellow color).

“As for T7 RNAP-mediated transcription, transcriptional activity is the major factor driving gene locus movement.”

2) As T7 RNA-polymerase doesn't couple with ribosomes, I'm confused as to why the kasugamycin experiment was performed with this strain and not the *E. coli* RNA-polymerase strain.

Performing this experiment with the *E. coli* RNA-polymerase would likely yield a much stronger affect on the localization to the nucleoid periphery if their interpretation that translation coupling is the most important factor is correct.

Answer: We thank Reviewer 2 for the comments and suggestion. We wanted to quantify the effect of the formation of a large DNA-RNAP-mRNA-ribosome complex on gene locus movement (the contribution of ribosome binding to mRNA to the degree of gene movement toward the nucleoid periphery). *E. coli* RNAP and ribosomes are known to physically interact (Burmam et al, *Science* **328**, 501-4, 2010; Demo et al, *Elife* **6**, 1-17, 2017; Fan et al, *Nucleic Acids Res.* **45**, 11043-11055, 2017; Kohler et al, *Science* **356**, 194-197, 2017). Thus, treatment with kasugamycin will inhibit translation initiation, but not the direct interactions between *E. coli* RNAP and the ribosome. Thus, we treated T7 RNAP-expressing cells with kasugamycin. As T7 RNAP does not interact with ribosomes, treatment with kasugamycin will inhibit the formation of the DNA-T7RNAP-mRNA-ribosome complex. We wanted to quantify this effect. As a result, we found that ribosome binding to nascent mRNA contributed to gene relocation by approximately 30-35% (Fig. 4d and e). We also confirmed this result by using ribosome binding site (RBS)-deleted strains (Fig. 4f) driven by T7 RNAP transcription.

As Reviewer 2 suggested, we investigated the effect of translation inhibition using kasugamycin treatment on gene movement in *E. coli* RNAP-driven transcription. We measured gene locus movement at 15 min after 5 mg/ml kasugamycin treatment using the *lacZ*-6×TetO strain (Fig. P6). We used tetR-mCrimson3 to detect gene loci and simultaneously examined nucleoid morphology using SYTOX Green dye (Bakshi et al, *Appl. Environ. Microbiol.* **80**, 4977-4986, 2014).

Inhibition of translation initiation using kasugamycin treatment resulted in no movement of the gene locus after transcription induction at 24 degrees (*lacZ*-6×TetO strain; Fig. P6a). This result is consistent with our result in the RBS-deletion strain (*lacZ*-6×TetO_dRBS), as shown in Fig. 3d of the original manuscript. We also confirmed that kasugamycin did not change the ratio of the nucleoid width relative to the cell width under our experimental conditions (Fig. P6b). Again, all these results support our initial conclusion that transcription-translation coupling is the major factor driving gene movement toward the nucleoid periphery.

Fig. P6. Effect of kasugamycin (KSG) on the movement of gene loci by *E. coli* RNAP-driven transcription.

(a) Average relative x-positions of gene loci were obtained in cells treated with 1 mM IPTG (induced) and without IPTG (repressed) at 15 min after 5 mg/ml kasugamycin treatment to the *lacZ-6xTetO* strain. The experiments were performed at 24 degrees. (b) The relative width of the nucleoid compared to the cell width with and without KSG treatment. The error bars were obtained from three independent experiments.

3) As RNase E is responsible for mRNA decay is localized to the inner membrane, and mRNA decay would break the transcription/translation coupling, driving an mRNA to the nucleoid periphery would potentially accelerate mRNA decay and reduce coupling. Constructs lacking the RBS also show lower extent at the periphery, but since its known that translation protects mRNA from decay, couldn't this effect be explained RNase E having enhanced cleavage due to the colocalization with the nascent mRNA? Fast mRNA decay has recently been shown for mRNAs for membrane proteins in *E. coli* that depends on this membrane tether for RNase E (Moffitt Elife 2016). Performing this experiment with a strain lacking the inner membrane tether for RNase E would be able to distinguish between translation and decay. This may be particularly important since the delta RBS constructs show increased mRNA decay rates.

Answer: We appreciate Reviewer 2 for commenting on an interesting point about RNase E.

As Reviewer 2 mentioned, movement of mRNA to the nucleoid periphery would increase the degradation of mRNA. However, because ribosomes are abundant near the plasma membrane, mRNAs must move toward the plasma membrane for efficient translation. As Reviewer 1 commented, the endcaps of *E. coli* are rich in 70S ribosomes. Thus, mRNAs should move to the ribosome-rich

cytosol. The detachment of mRNAs from DNA (the loss of transcription-translation coupling) after the movement of the DNA-RNAP-mRNA-ribosome complex to the nucleoid periphery will happen for the efficient translation of mRNAs. Of course, competition between translation and mRNA decay will occur here. Regarding mRNA degradation, RNase E is incredibly efficient. However, some exonucleases (RNase R or PNPase) and endonucleases, such as RNase G, RNase Z, RNase III, or others, are not anchored to the inner membrane and are able to degrade RBS-deleted mRNAs (Mohanty & Kushner, *Annu. Rev. Microbiol.* **70**, 25-44, 2016). Therefore, RBS-deleted mRNA can be degraded in the area around the nucleoid by these RNases.

Fig. P7. Cartoon comparing the sizes of the cell and nucleoid and showing the location of the *lacZ* gene.

It is to be noted that ribosomes are freely moving in the cytosol, while RNase E is anchored on the inner membrane. As we presented in Fig. 2c, the *lacZ* gene is most often found at the center of the nucleoid. The physical distance between the center of the nucleoid and inner membrane is 0.5 μm (500 nm). Fig. P7 shows the sizes of the cell and nucleoid and the location of the *lacZ* gene locus for the purpose of comparison. We calculated the size of 4.5 kb mRNA using a freely jointed chain model (Chen et al, *Proc. Natl. Acad. Sci. U S A* **109**, 799-804, 2012; Leija-Martínez et al, *Nucleic Acids Res.* **42**, 13963-13968, 2014). The size of the mRNA was 80 nm (radius of gyration = 40 nm), even without considering the formation of secondary structure. Because RNase E is anchored on the inner membrane, the distance between RNase E and the mRNA generated at the center of the nucleoid is 500 nm. Thus, if the interaction between RNase E and mRNA is responsible for movement of gene loci, this movement is biased to the nucleoid periphery. This bias means that the gene locus at the

central region of the nucleoid has a very low chance of moving, while mRNAs generated at the nucleoid periphery are more likely to interact with RNase E. However, as shown in Fig. 2d, a simulation assuming random movement of gene loci recapitulated the experimental data quite well. If we assume biased movement in the simulation, the gene loci distribution after induction adopts a very different shape compared with that of our data in Fig. 2d.

In addition, the additional experimental results in this revised manuscript strongly support that transcription-translation coupling is the major factor driving DNA loci movement toward the nucleoid periphery in *E. coli* RNAP transcription. As Reviewer 2 suggested, we performed two additional experiments: the treatment to *E. coli* RNAP transcription with kasugamycin (see the answer to question 2 from Reviewer 2) and inhibiting translation by developing a strain lacking a start codon (see the response to other comments from Reviewer 2). In particular, when we blocked translation by deleting both the RBS and start codon, no gene movement was observed at 37 degrees (Fig. P8). This result strongly emphasizes that transcription-translation coupling is the major factor driving the observed gene locus movement.

Fig. P8. Translation plays a major role in the movement of DNA loci toward the nucleoid periphery in *E. coli* RNAP transcription. We constructed the lacZ-12×TetO, lacZ-12×TetO_ΔRBS, and lacZ-

12×TetO_ΔRBS_ΔATG strains. The gene location was observed using TetR-mCrimson3, and nucleoid morphology was simultaneously imaged using SYTOX Green dye. (a) 5' UTR sequence of the mutated strains. The underlined AGGAAA in the WT strain was mutated to TCTCTC (red letters) (ΔRBS) for RBS substitution. The start codon of the *lacZ* gene (underlined) was mutated to TAA (ΔRBSΔATG). (b) β-Galactosidase assay showing that LacZ expression in the strain in which the start codon in the ΔRBS strain was replaced by a stop codon (ΔRBSΔATG) decreased to approximately 20% of that in the ΔRBS strain (blue bars, 1 mM IPTG induction of each strain). The LacZ expression level in ΔRBSΔATG cells was lower than that in WT cells under repressed conditions (no IPTG, gray bar) (c) Average x-positions of the *lacZ* gene locus were determined in the absence (repressed, gray bars) and presence of IPTG (induced, blue bars). (d) The degree of gene movement is the difference in the average position before (repressed) and after (induced) induction in (c). Error bars were obtained from two or three independent experiments.

Other comments:

As the delta RBS lowers translation, but doesn't completely abolish it, wouldn't a stronger effect be observed from a start codon deletion?

Answer: We thank Reviewer 2 for the excellent suggestion. As Reviewer 2 suggested, we constructed a strain in which the start codon of the *lacZ* gene is replaced by a stop codon in the ΔRBS strain (lacZ-12×TetO_ΔRBSΔATG) (Fig. P8a). As Reviewer 2 expected, the strain exhibited a stronger inhibitory effect on translation than that in the RBS-deleted strain (Fig. P8b). In addition, the movement of gene loci in the lacZ-12×TetO_ΔRBSΔATG strain was nearly eliminated; there was no difference in the average gene locus position before and after induction (Fig. P8c and d). As a result, the gene movement distance is strongly correlated with translation activity (Fig. P8d). These results strongly support our initial conclusion that transcription-translation coupling is a major factor driving the movement of DNA loci to the nucleoid periphery in *E. coli* RNAP transcription.

This new experimental result from an experiment suggested by Reviewer 2 strongly supports our initial conclusion. We added the results shown in Fig. P8 to the revised manuscript (Supplementary Fig. 10). Accordingly, we added the following sentences to the main text (page 8, highlighted in yellow color).

“Then, to completely abolish translation, we removed the start codon from the lacZ mRNA (Supplementary Fig. 10a) and found that the strain showed stronger translation inhibition than

that in the RBS-deleted strain (Supplementary Fig. 10b). As expected from the decreased translation activity, we did not observe gene movement in the strain in which the start codon from the lacZ mRNA had been deleted (Supplementary Fig. 10c). The degree of gene movement was strongly correlated with translation activity (Supplementary Fig. 10d). These results show that the coupling of transcription with translation is the major factor determining the subcellular relocation of gene loci under transcription in *E. coli*.”

Change "dots" to foci, as this is more common terminology.

Answer: We thank Reviewer 2 for this comment. We replaced ‘dots’ with ‘foci’ in the revised manuscript.

The manuscript would be improved by significant improvement in english grammar.

Answer: We have received another round of commercial English editing from Nature Research Editing Service for this revised manuscript.

on line 144 what are the units for 0.07?

Answer: We used the relative coordinates showing localization of the gene locus in a cell; the spot position was normalized by the cell radius. Thus, 0.07 has no unit. We explain this clearly in the revised manuscript.

Reviewers' comments:

Reviewer #1 (Remarks to the Author):

The revised manuscript is excellent and well worth publishing. I believe the authors have addressed all the main concerns in the first review. I recommend publication without change.

Reviewer #2 (Remarks to the Author):

Overall this paper by Yang et al. claims to directly observe that transcription-translation coupling relocates an actively transcribed DNA locus to the nucleoid periphery in *E. coli* cells. More strongly, this paper shows that transcription, whether by T7 RNAP or *E. coli* RNAP, promotes relocation of the lac-operon loci to the nucleoid periphery. Importantly, this occurs on an mRNA which does not encode a membrane protein. While transcription-translation coupling is the focus of the writing, few experiments in the main figures of the paper support this claim of "transcription-translation coupling" being the main factor for gene locus relocation. In fact transcription-translation coupling occurs only between the *E. coli* RNAP and ribosomes and not T7, yet only one main figure of the paper contains *E. coli* RNAP and T7 is used for all subsequent experiments. New experiments performed after the last review significantly improve the claim of a "direct observation" presented in the title, but these data were either put in the supplement (Fig S10) or not present in the paper at all (Fig P6). Overall I find the data of high quality, but the writing of this draft is only minimally improved. This draft is still presented in a confusing way that muddles the beautiful data they have collected. However, I believe that with significant rewriting, the paper could be very good. In the results presented, the amount of transcription observed (regardless of polymerase) directly correlates with the movement of DNA to the periphery, and this to me is the strongest result.

The title claims that the authors directly observe gene-locus movement through transcription-translation coupling. However, the authors do not directly address transcription-translation coupling. Instead the authors indirectly address the role of translation in loci movement by making RBS mutations, start codon mutations, and add the drug Kasugamycin. While all of these disrupt translation initiation, they do not directly probe transcription-translation coupling. Perhaps if the authors used FRET between RNAP and the ribosome together with loci tracking they could claim this, but currently its inferred indirection from mutations in the mRNA. Little effort was performed in the interpretation of the results to ensure other processes are not the cause of this lack of loci movement (such as enhanced mRNA degradation in the delta RBS strain which the authors do observe in figure S15). The language throughout the manuscript relating to transcription-translation coupling should be toned down to account for these uncertainties. In many cases, the interpretation of the results appear to be exaggerated. For example on page 9 line 214, "These results show that the coupling of transcription with translation is the major factor determining the subcellular relocation of gene loci under transcription in *E. coli*" and even the "Direct observation" in the title.

The introduction fails to introduce background information on T7 RNAP. Background on T7 would be critical to introduce since all but one data figures in the main text focus on T7 RNAP. Why was T7 used for some experiments, and *E. coli* RNAP used for others. This justification should be clearer through the manuscript. The discussion also fails to adequately discuss and put in context the T7 RNAP results.

Reviewers' comments:

Reviewer #1 (Remarks to the Author):

The revised manuscript is excellent and well worth publishing. I believe the authors have addressed all the main concerns in the first review. I recommend publication without change.

Answer: We appreciate Reviewer 1's positive evaluation.

Reviewer #2 (Remarks to the Author):

Overall this paper by Yang et al. claims to directly observe that transcription-translation coupling relocates an actively transcribed DNA locus to the nucleoid periphery in *E. coli* cells. More strongly, this paper shows that transcription, whether by T7 RNAP or *E. coli* RNAP, promotes relocation of the lac-operon loci to the nucleoid periphery. Importantly, this occurs on an mRNA which does not encode a membrane protein. While transcription-translation coupling is the focus of the writing, few experiments in the main figures of the paper support this claim of "transcription-translation coupling" being the main factor for gene locus relocation. In fact transcription-translation coupling occurs only between the *E. coli* RNAP and ribosomes and not T7, yet only one main figure of the paper contains *E. coli* RNAP and T7 is used for all subsequent experiments. New experiments performed after the last review significantly improve the claim of a "direct observation" presented in the title, but these data were either put in the supplement (Fig S10) or not present in the paper at all (Fig P6). Overall I find the data of high quality, but the writing of this draft is only minimally improved. This draft is still presented in a confusing way that muddles the beautiful data they have collected. However, I believe that with significant rewriting, the paper could be very good. In the results presented, the amount of transcription observed (regardless of polymerase) directly correlates with the movement of DNA to the periphery, and this to me is the strongest result.

Answer: We appreciate Reviewer 2's positive comments and suggestions.

According to Reviewer 2's comments, we moved Fig. S10 and Fig. P6 to the main figures (Fig. 4) and the supplementary figures (Supplementary Figure 10), respectively, and modified the text according to the change of the figures in the re-revised manuscript.

The title claims that the authors directly observe gene-locus movement through transcription-translation coupling. However, the authors do not directly address transcription-translation coupling. Instead the authors indirectly address the role of translation in loci movement by making RBS mutations, start codon mutations, and add the drug Kasugamycin. While all of these disrupt translation initiation, they do not directly probe transcription-translation coupling. Perhaps if the authors used FRET between RNAP and the ribosome together with loci tracking they could claim this, but currently it's inferred indirectly from mutations in the mRNA. Little effort was performed in the interpretation of the results to ensure other processes are not the cause of this lack of loci movement (such as enhanced mRNA degradation in the delta RBS strain which the authors do observe in figure S15). The language throughout the manuscript relating to transcription-translation coupling should be toned down to account for these uncertainties. In many cases, the interpretation of the results appear

to be exaggerated. For example on page 9 line 214, "These results show that the coupling of transcription with translation is the major factor determining the subcellular relocation of gene loci under transcription in *E. coli*" and even the "Direct observation" in the title.

Answer: We appreciate Reviewer 2's comments and suggestions. We modified the manuscript in order to tone down our interpretation as Reviewer 2 suggested.

In Title:

According to Reviewer 2's comments, we deleted "Direct" from the title.

In Introduction section:

"Among these factors, transcription-translation coupling is the major factor determining protein gene locus movement in *E. coli* RNAP transcription." → "Our findings show that transcription-translation coupling significantly enhances the gene locus movement in *E. coli* RNAP transcription."

In Results section:

"These results show that the coupling of transcription with translation is the major factor determining the subcellular relocation of gene loci under transcription in *E. coli*" → "These results show that the coupling of transcription with translation is an important factor for the subcellular relocation of gene loci under transcription in *E. coli*."

In Discussion section:

"transcription-translation coupling contributes most significantly to gene locus movement in *E. coli* RNAP-driven transcription." → "transcription-translation coupling contributes significantly to gene locus movement in *E. coli* RNAP-driven transcription."

The introduction fails to introduce background information on T7 RNAP. Background on T7 would be critical to introduce since all but one data figures in the main text focus on T7 RNAP. Why was T7 used for some experiments, and *E. coli* RNAP used for others. This justification should be clearer through the manuscript. The discussion also fails to adequately discuss and put in context the T7 RNAP results.

Answer: We appreciate Reviewer 2's comments and suggestions. According to Reviewer 2's comments, we modified our manuscript.

1. We included the background information why we used T7 RNAP in **Introduction** section (highlighted in yellow).
2. We clarified the reason we used T7 RNAP and *E. coli* RNAP in each section.

Localization of gene loci actively transcribed by T7 RNAP section: "In order to visualize the location of a specific gene locus occupied by RNAPs, we replaced the endogenous promoter of the *lac* operon with a T7 RNAP-specific promoter (Fig. 1a)."

Movement of a non-membrane protein gene loci by *E. coli* RNAP section: "Our observation that transcription by T7 RNAP causes the movement of gene loci immediately raises the question

of whether this movement also occurs with transcription by endogenous E. coli RNAP.”

Effect of transcription-translation coupling on gene loci movement section: “We used E. coli RNAP-driven transcription in this study because transcription by E. coli RNAP is coupled with translation while T7 RNAP transcription is decoupled with translation.”

Gene loci movement depends on the transcriptional activity section: “The T7 RNAP transcription system allows us to control the transcriptional activity through T7 RNAP concentration variation. Thus, we used T7 RNAP transcription system in this study.”

Analysis of the factors contributing to gene loci movement section: “To remove the effects of direct interactions between RNAPs and ribosomes¹⁵⁻¹⁷ and quantify only the effect of ribosome binding to mRNAs to the degree of gene loci movement, we used T7 RNAP-driven transcription for this analysis (T7p_lacZ-12×TetO). T7 RNAP has no known interaction with ribosomes and moves approximately five times faster than E. coli RNAP.”

3. We put the results of T7 RNAP in **Discussion** section.

“Our work shows that the major factor that causes the gene loci movement during transcription is different for T7 RNAP- and E. coli RNAP-driven transcription. As for E. coli RNAP-driven transcription, the coupling between transcription and translation causes most gene loci movement that the deletion of both RBS site and a start codon (decoupling between transcription and translation) nearly abolishes the gene loci movement (Fig. 3 and Fig. 4). However, the transcription by T7 RNAP is not coupled with translation. The deletion of RBS site causes only 30% less movement of gene loci (Fig. 5f), which means that ribosome binding to mRNA increases the gene loci movement only 30% more. Thus, as for T7 RNAP-driven transcription, the transcriptional activity causes most gene loci movement.”